# TAZ Controls *Helicobacter pylori*-Induced Epithelial–Mesenchymal Transition and Cancer Stem Cell-Like Invasive and Tumorigenic Properties

**DOI:** 10.3390/cells9061462

**Published:** 2020-06-13

**Authors:** Camille Tiffon, Julie Giraud, Silvia Elena Molina-Castro, Sara Peru, Lornella Seeneevassen, Elodie Sifré, Cathy Staedel, Emilie Bessède, Pierre Dubus, Francis Mégraud, Philippe Lehours, Océane C.B. Martin, Christine Varon

**Affiliations:** 1INSERM, UMR1053 Bordeaux Research in Translational Oncology, BaRITOn, University of Bordeaux, F33000 Bordeaux, France; camille.tiffon@u-bordeaux.fr (C.T.); julie.giraud085@gmail.com (J.G.); smolina13@gmail.com (S.E.M.-C.); sara.peru@etu.u-bordeaux.fr (S.P.); lornella.seeneevassen@u-bordeaux.fr (L.S.); elodie.sifre@u-bordeaux.fr (E.S.); emilie.bessede@u-bordeaux.fr (E.B.); pierre.dubus@u-bordeaux.fr (P.D.); francis.megraud@chu-bordeaux.fr (F.M.); philippe.lehours@u-bordeaux.fr (P.L.); oceane.martin@u-bordeaux.fr (O.C.B.M.); 2Institute for Health Research, University of Costa Rica, Sabanilla 11502, Costa Rica; 3INSERM, UMR1212, University of Bordeaux, F-33000 Bordeaux, France; Cathy.staedel@inserm.fr; 4CHU de Bordeaux, F-33000 Bordeaux, France

**Keywords:** gastric cancer, *Helicobacter pylori*, hippo pathway, ZEB1, epithelial–mesenchymal transition, TAZ, WWTR1, cancer stem cells, YAP

## Abstract

*Helicobacter pylori* infection, the main risk factor for gastric cancer (GC), leads to an epithelial–mesenchymal transition (EMT) of gastric epithelium contributing to gastric cancer stem cell (CSC) emergence. The Hippo pathway effectors yes-associated protein (YAP) and transcriptional co-activator with PDZ binding motif (TAZ) control cancer initiation and progression in many cancers including GC. Here, we investigated the role of TAZ in the early steps of *H. pylori*-mediated gastric carcinogenesis. TAZ implication in EMT, invasion, and CSC-related tumorigenic properties were evaluated in three gastric epithelial cell lines infected by *H. pylori*. We showed that *H. pylori* infection increased TAZ nuclear expression and transcriptional enhancer TEA domain (TEAD) transcription factors transcriptional activity. Nuclear TAZ and zinc finger E-box-binding homeobox 1 (ZEB1) were co-overexpressed in cells harboring a mesenchymal phenotype in vitro, and in areas of regenerative hyperplasia in gastric mucosa of *H. pylori*-infected patients and experimentally infected mice, as well as at the invasive front of gastric carcinoma. TAZ silencing reduced ZEB1 expression and EMT phenotype, and strongly inhibited invasion and tumorsphere formation induced by *H. pylori*. In conclusion, TAZ activation in response to *H. pylori* infection contributes to *H. pylori*-induced EMT, invasion, and CSC-like tumorigenic properties. TAZ overexpression in *H. pylori*-induced pre-neoplastic lesions and in GC could therefore constitute a biomarker of early transformation in gastric carcinogenesis.

## 1. Introduction

Gastric cancer (GC) is the third leading cause of cancer-related death worldwide [1,2]. GCs are mostly non-cardia gastric adenocarcinomas, occurring in patients around 70 years old. In most situations, GC has an infectious origin, attributed to the infection by the bacterium *Helicobacter pylori*. Half of the world’s population is infected with *H. pylori*, and this infection is detected in more than 92% GC cases [3,4].

Infection with *H. pylori* always leads to chronic inflammation of the gastric mucosa, which can potentially evolve slowly into atrophy, metaplasia, and dysplasia, and in the worst scenario leads to non-cardia gastric carcinoma after several decades [5]. The major *H. pylori* virulence factor is carried by the *cag* pathogenicity island (*cag*PAI), which encodes a type 4 secretion system (T4SS) and the cytotoxin associated gene A (*cag*A) oncoprotein (CagA). When CagA is injected into gastric epithelial cells via the T4SS, it can induce perturbations of several signaling pathways, leading to the destabilization of inter-cellular junctions, polarity complexes, and pro-inflammatory responses [6]. Previous studies, including those performed in our laboratory, have shown that *H. pylori* induces an epithelial–mesenchymal transition (EMT) of epithelial cells of the gastric mucosa. EMT is a very well-known pathophysiological trans-differentiation process that confers mesenchymal phenotype and properties to epithelial cells. In the gastric context, this EMT is characterized by the loss of epithelial polarity and cellular junctions and the acquisition of a mesenchymal, motile phenotype called the “hummingbird” phenotype [7,8,9,10]. The overexpression of zinc finger E-box-binding homeobox 1 (ZEB1) and Snail transcription factors and of structural components such as Vimentin, as well as migration and invasion capacities are reminiscent events of the EMT process. EMT also occurs during cancer dissemination to allow cell extravasation through blood vessels and dissemination to distant organs, thereby initiating metastases [11]. EMT can also lead to the emergence of cells with cancer stem cell (CSC) properties in different cancers including GC [12,13,14]. CSCs represent a rare cell subpopulation within the tumor that is able to initiate tumor development and dissemination to form distant metastases. CSCs are more resistant to conventional chemotherapy than the more differentiated tumor cells and can be identified by the expression of immaturity markers such as cluster of differentiation 44 (CD44) and aldehyde dehydrogenase 1 family member A1 (ALDH1A1) in GC [15,16,17]. Their recent discovery in GC [15,17,18,19] is a very promising research axis, allowing an earlier detection of the cells at the origin of CSC in pre-neoplastic lesions, as well as the development of CSC-based targeted therapies [20,21].

Several pathways, including the Hippo signaling pathway, have been described to control CSC properties. The Hippo pathway, a highly conserved signaling pathway, from fruits flies to humans, is involved in physiology in the modulation of organ size during development and the maintenance of stemness, especially in the gastrointestinal tract. Its dysregulation, in pathological conditions, can lead to cancer emergence and progression [22,23,24,25]. The Hippo pathway is controlled by upstream regulators that activate a module of inhibitory kinases, which in turn inhibits a transducer module composed of oncogenic co-transcription factors. Upstream regulators involve components of cell/cell junctions, polarity complexes, and extracellular matrix stiffness, all acting on the regulation of the inhibitory kinases, including two serine/threonine kinases: Mammalian sterile 20-like kinase-1/2 (MST1/2) and its target the large tumor suppressor kinase 1/2 (LATS1/2). When the Hippo pathway is activated, LATS1/2 is phosphorylated, which in turn phosphorylates its downstream targets yes-associated protein (YAP) and transcriptional co-activator with PDZ binding motif (TAZ) on serine residues, resulting in their sequestration in the cytoplasm and subsequent degradation by the proteasome [25,26,27,28]. When the Hippo pathway is inactivated, YAP and TAZ are not phosphorylated by LATS1/2 and can therefore accumulate in the nucleus and bind to transcription factors such as the TEA domain (TEAD) transcription factor family members, their main partners. The resulting complexes activate transcriptional programs inducing cellular plasticity, proliferation, or drug resistance [29].

Recent work from our laboratory showed that the Hippo kinase LATS2 controls *H. pylori*-induced EMT and intestinal metaplasia [30]. YAP1/TEAD-mediated transcription is activated first following *H. pylori* infection and then repressed later while LATS2 accumulates. LATS2 appears to be a protective factor, limiting the loss of gastric epithelial cell identity that normally precedes neoplastic transformation and GC development. The role of YAP has been widely demonstrated in cancer initiation and progression [25,26,27], including GC [31,32,33]. Its paralogue TAZ has also been implicated in aggressiveness and metastasis in different cancers [34,35,36,37,38,39] and recent literature shows its involvement in GC aggressiveness, metastasis, and CSC properties [40,41,42]. In GC xenograft models, inhibition of YAP/TAZ interaction with TEADs by the pharmacological inhibitor verteporfin inhibits the tumorigenic properties of CSCs in GC [43]. TAZ is overexpressed in 66.4% GC [40], in which its overexpression is correlated with lymphatic metastasis and tumor stage [44]. In GC cell lines, studies have shown that TAZ controls cell migration, and its overexpression is associated with EMT [40,42]. Until now, the role of TAZ has never been investigated in response to *H. pylori* infection in the context of EMT and early steps of gastric carcinogenesis.

This study aimed to highlight TAZ implication in *H. pylori*-induced EMT and the gastric carcinogenesis process, both in vitro and ex vivo. For this purpose, we explored TAZ regulation in gastric epithelial cells in response to *H. pylori* infection and the consequences of its inhibition by interference RNA strategies on *H. pylori*-induced EMT and CSC-like invasive and tumorigenic properties. Furthermore, the expression of both TAZ and the major EMT marker ZEB1 were evaluated in GC patient-derived tissues and mouse models.

## 2. Materials and Methods

### 2.1. Cell and Bacterial Culture and Co-Culture

MKN45 (RIKEN, BRC Cell Engineering Division, Japan) and MKN74 (HSRR Bank JCRB0255) were cultivated in Roswell park memorial institute (RPMI) 1640 Glutamax medium (Thermo Fisher Scientific, Courtaboeuf, France), and NCI-N87 (ATCC CRL-5822), Kato-III (ATCC HTB-103), and GC07 were cultivated in Dulbecco’s modified eagle medium (DMEM-F12) medium (Thermo Fisher Scientific), as previously described [12,15,45]. GC07 was derived from a patient-derived GC xenograft (PDX), which was successfully established by serial transplantation in non-obese diabetous severe combined immunodeficiency Interleukin 2 gamma null (NSG) mice, as previously described [15]. All cells were grown in medium supplemented with 10% heat-inactivated fetal bovine serum (Eurobio, Les Ulis, France) and 50 µg/mL vancomycin (Sandoz, Holzkirchen, Germany) at 37 °C in a 5% CO_2_ humidified atmosphere. The experiments were performed with cells seeded at low density (3 × 10^4^ cells per well in 24 well plates). Phase contrast microscopy images of cell cultures were taken using the ZOE fluorescent Cell Imager (BioRad, Marnes-la-Coquette, France).

The *cag*A-positive 7.13 wild type (WT) *H. pylori* strain was used for most of co-culture experiments. Several *H. pylori* strains were used: 7.13 WT strain and its isogenic mutant deleted for *cagA* (Δ*cag*A) or *cagE* (Δ*cag*E) (from Richard Peek, Vanderbilt University Medical Center, Nashville, TN, USA) and the P12 WT strain and its isogenic mutant deleted for *cag*A (Δ*cag*A) or the *cag*PAI (Δ*cag*PAI) (from Rainer Haas, Ludwig Maximilian University of Munich, Munich, Germany). *H. pylori* were cultivated on Columbia agar plates supplemented with 10% human blood under microaerophilic (5% O_2_) conditions at 37 °C in humidified atmosphere, as previously described [8,13]. Co-culture experiments were performed at a multiplicity of infection of 1:25. The evaluation of bacterial number was estimated by spectrophotometry with OD_600nm_ = 1 corresponding to 2 × 10^8^ bacteria/mL. DNA was extracted for gastric tissues and PCR were performed for the detection of *H. pylori* and *cag*A, as previously described [46].

### 2.2. Transfection of siRNA

In order to downregulate TAZ expression, we transfected cells with small interfering RNA (siRNA) designed to target human TAZ. The siRNA suspension was composed of Lipofectamine RNAiMAX (Thermo Fisher Scientific), OptiMEM medium (Thermo Fisher Scientific), and 20 nM of siRNA. Experiments were performed with three different siRNA that target TAZ: siTAZ1 (5′- ACGUUGACUUAGGAACUUU-3′, Eurofins, Dortmund, Germany) [47], siTAZ2 (5′-AGGUACCUUCCUCAAUACA-3′, Eurofins) [47], and siTAZ3 (5′-UGUGGAUGAGAUGGAUACA-3′, Eurofins). siTAZ1 and siTAZ2 are specific of *TAZ* sequences. siTAZ3 recognizes a common sequence of both TAZ and YAP mRNA. A negative control siCtrl (siCtrl 5′-GGGCAAGACGAGCGGGAAG-3′, Eurofins) with no known homology to mammalian genes was used. Two rounds of siRNA transfection were performed (separated by 24 h) in medium without antibiotics. Co-culture experiments were performed 6 h after the last transfection. Analyses were performed 30 h after the last transfection and 24 h after infection [13,30]. TAZ inhibition efficiency was evaluated by Western blot experiments.

### 2.3. TEAD-Luciferase Reporter Assay

TEAD reporter assay was performed by co-transfecting the *Renilla* luciferase reporter (pRL-SV40, Promega) and either the TEAD reporter (8xGTIIC-Luciferase reporter, gift from Stefano Piccolo [25]) or TATA box control (TAL, Becton Dickinson, Le Pont de Claix, France) firefly luciferase reporters to evaluate the relative TEAD activity. The transfection was performed using Lipofectamine 2000 (Thermo Fisher Scientific) and 12.5 ng of *Renilla* vector, 125 ng of pTA-Luciferase vector, or 125 ng of pTEAD-Luciferase vector. After 24 h, cells were lyzed and the luminescence quantification was measured using the Dual-Luciferase Reporter Assay system according to the manufacturer’s recommendations (Promega, Charbonnières-les-bains, France). Firefly luciferase activities were normalized for transfection efficiency by *Renilla* luciferase for each sample, and TEAD-specific luciferase activity was then normalized to TAL activity.
$$((\frac{Luciferase}{Renilla})\;\text{pcDNA}\;\text{TEAD}\;\text{8xGTIIC-Luciferase}/((\frac{Luciferase}{Renilla})\;\text{pcDNA}\;\text{pTA-Luciferase}).$$

### 2.4. RNA Extraction and RTqPCR

Total RNA was extracted using TRIZOL reagent (Thermo Fisher Scientific) according to the manufacturer’s recommendations and quantified by their absorbance at 260 nm. Reverse transcription was performed using the QuantiTect Reverse Transcription kit (Qiagen, Courtaboeuf, France) containing a DNA digestion step. Quantitative PCR was performed using specific primers at 0.3 µM and the SYBR-PCR-Premix Ex-Taq reagent (Thermo Fisher Scientific) (Table 1). The amplification steps contain 15 s at 94 °C, 30 s at 60 °C, and 30 s at 72 °C for 45 cycles. Relative expressions were calculated using the comparative cycle threshold (Ct) method with hypoxanthine phosphoribosyltransferase 1 (*HPRT1)* and TATA-Box binding protein (*TBP)* as normalizers. Primers used were: Cysteine-rich angiogenic inducer 61 (*CYR61*), Matrix metallopeptidase 9 (*MMP9*), Snail family transcriptional repressor 1 (*SNAI1*), Vimentin (*VIM*), and *ZEB1*.

### 2.5. Western Blot

After having been washed twice with phosphate-buffered saline (PBS), cells were lysed on ice in ProteoJET reagent (Thermo Fisher Scientific) supplemented with protease/phosphatase inhibitor cocktail (Sigma-Aldrich, St. Quentin Fallavier, France). The extracted proteins were submitted to SDS-polyacrylamide gel electrophoresis (BioRad, Marnes-la-Coquette, France) and Western blotting on nitrocellulose membranes (BioRad) for immuno-labelling. After 1 h in bovine serum albumin (BSA)-blocking solution (5%), immuno-labelling was performed using mouse anti-TAZ (1:2000, cat. 560235, BD Pharmingen, Thermo Fisher Scientific), rabbit anti-YAP1 (1:2000; cat. 4912 Santa Cruz Biotechnology, Heidelberg, Germany), mouse anti-Glyceraldehyde 3 phosphate dehydrogenase (GAPDH) (1:2000; sc-47724 Santa Cruz Biotechnology), and mouse anti-α-tubulin (1:8000; cat. T-6074, Sigma-Aldrich) for 1–2 h at room temperature. Fluorescent-labelled secondary antibodies (1:2000; StarBright 700 goat anti-mouse cat. 12004158 or StarBright 700 goat anti-rabbit 12004161, BioRad) were used to detect the primary antibodies and were detected by fluorescence imaging with the BioRad Chemidoc imager. Band intensities were normalized to GAPDH or α-tubulin, and were quantified using the ImageJ software (National Institutes of Health, Bethesda, MD, USA) [48].

### 2.6. Immunofluorescence and Quantification of the Stainings

Experiments were performed as previously described [13,30] on glass coverslips at a density of 2.5 × 10^4^ cells per well. Cells were fixed in a 4% paraformaldehyde solution in PBS supplemented with 1 mM CaCl_2_ and 1 mM MgCl_2_ for 10 min, and permeabilized in 0.1% Triton in Tris-buffered saline (TBS) supplemented with 1 mM CaCl_2_ and 1 mM MgCl_2_ for 1 min. After blocking with 1% bovine serum albumin and 2% fetal calf serum solution in TBS, cells were incubated with the primary antibodies mouse anti-TAZ (1:100, cat 560235, BD Pharmingen) and rabbit anti-ZEB1 (1:100, cat. A301-922, Bethyl, Souffelweyersheim, France) for 1 h at room temperature. After 3 washes in PBS solution, fluorescent-labelled secondary antibodies were incubated for 1 h at room temperature using either donkey anti-mouse Alexa-488-labelled (1:250, cat. A21202 Thermo Fisher Scientific), goat anti-rabbit Alexa-546-labelled (1:250, cat. A11010 Thermo Fisher Scientific), or goat anti-rabbit Alexa-488-labelled (1:250, cat. A32731, Thermo Fisher Scientific) secondary antibodies mixed with either Alexa-546-labelled phalloidin (1:250, cat. A22283, Thermo Fisher Scientific) or Alexa-647-labelled phalloidin (1:200, cat. A222287, Thermo Fisher Scientific) and 4′-6-diamidino-phenyl-indol (DAPI) (50 µg/mL, cat. D9542, Sigma-Aldrich). After being washed 3 times in PBS and having 1 last wash in distilled water, the coverslips were mounted on glass slides using Slowfade reagent (cat. S36936, Thermo Fisher Scientific). Images were taken using a ×40 (numerical aperture, 1.3) oil immersion objective on an Eclipse 50i epi-fluorescence microscope (Nikon, Champigny sur Marne, France) with Nis Element acquisition software. Fiji software was used for the relative quantification of the signal intensity on digital images [48]. The relative amount of TAZ and ZEB1 nuclear accumulation was quantified in the nucleus of each cells by delimiting nuclei using DAPI staining and quantifying the mean fluorescence intensity (pixels shade of grey) of TAZ and ZEB1 labelling with Fiji software [48]. In some experiments, the number of positive- vs. negative-labelled nuclei for MKN45 cells was determined. Nuclei were considered positive when the mean fluorescence intensity was higher than the mean fluorescence intensity of the uninfected cells plus two standard deviations.

### 2.7. Tumorsphere Assay

After 24 h of *H. pylori* infection and 30 h after the last siRNA transfection, we seeded 100 cells (MKN45 and NCI-N87) or 1000 cells (GC07) in non-adherent 96-well culture plates that were previously coated with a 10% poly-2 hydroxyethyl methacrylate (Sigma-Aldrich) solution in 95% (*v*/*v*) ethanol and dried overnight at 56°C. The medium composition is serum-free GlutaMAX-DMEM/F12 medium supplemented with 20 ng/mL of epidermal growth factor, 20 ng/mL of basic-fibroblast growth factor, 0.3% glucose, 5 µg/mL of insulin, and 1:100 N_2_ supplement (all from Thermo Fisher Scientific and Sigma Aldrich), and cultured at 37°C in a 5% CO_2_ humidified atmosphere [13,43]. Tumorspheres were counted 7 days after seeding under non-adherent conditions. 

### 2.8. Invasion Assay System

After 24 h *H. pylori* infection and 30 h siRNA-transfection, we recovered cells by trypsinization, and 3 × 10^4^ cells per condition were seeded in the upper side of 8 µm pored Transwells (Sigma-Aldrich) previously coated with rat-tail type 1 collagen (Becton Dickinson) [13]. After they were incubated for 18 h at 37 °C, inserts were fixed in 4% paraformaldehyde-PBS solution and labelled with DAPI. Cells from the upper part of the Transwell inserts were removed by swabbing. Cells that had invaded through the lower side of the inserts were counted using the ZOE fluorescent Cell Imager (BioRad).

### 2.9. Immunohistochemistry

Tissue sections that were 3 µm thick were prepared from formalin-fixed, paraffin-embedded human and mice tissues and were processed for immunohistochemistry procedures as previously described [12,30]. Primary antibodies rabbit anti-TAZ (1:80 for 1 h incubation, Sigma-Aldrich; HPA007415), rabbit anti-ZEB1 (1:800 during 2 h incubation for mice tissues and 1:100 during 2 h incubation for human tissues, Bethyl; A301-922A), mouse anti-Ki67 (1:150 during 30 min incubation for mice tissues, 652,401 clone 16A8, BioLegend, San Diego, CA, USA), or mouse anti-Ki67 (1:75 during 30 min incubation for human tissues, M7240 clone mib-1, Dako, Les Ulis, France). Secondary horseradish peroxidase (HRP)-labelled ImmPRESS anti-rabbit or anti-mouse antibodies for TAZ and ZEB1, ImmPRESS anti-rat for Ki67 from BioLegend, and VECTASTAIN Elite avidin-biotin complex (ABC) HRP kit (Peroxidase, Universal) for human Ki67 from Dako were incubated for 30 min. Immunolabeling was revealed after a 1–10 min incubation in liquid diaminobenzidine-chromogen substrate. Slides were counterstained with hematoxylin, dehydrated, and mounted with Eukitt-mounting medium (VWR, Fontenay-sous-Bois, France). Quantification scores of the relative expression of ZEB1, Ki67, and nuclear TAZ in gastric epithelial cells were determined in a blind lecture using a scale from 0 to 4 with the following criteria: 0: no staining, 1: <5% of positive epithelial cells; 2: 5 to <25% of positive cells; 3: 25 to <50% of positive cells; 4: ≥50% of positive epithelial cells.

### 2.10. Ethic Statements on Human and Mouse Tissue Samples

Studies on paraffin-embedded tissue samples from gastric adenocarcinoma patients were performed in agreement with the Direction for Clinical Research and the Tumour and Cell Bank of the University Hospital Centre of Bordeaux (Haut-Leveque Hospital, Pessac, France), as previously reported [13,15]. Animal experiments were performed in level 2 animal facilities of the University of Bordeaux (France), in conformity with the French Ministry of Agriculture Guidelines on Animal Care and the French Committee of Genetic Engineering and with the approval of institutional guidelines determined by the local Ethical Committee of the University of Bordeaux (approval number 4608), as described previously [30].

### 2.11. Statistical Analysis

Quantification values represent the mean of three or more independent experiments, each performed in duplicate or more ± standard error of the mean (SEM). Kruskal–Wallis test with Dunn’s post-test or ANOVA with Dunnett’s post-test were used to compare multiple groups. Statistics were performed using GraphPad Prism 8 software (San Diego, CA, USA).

## 3. Results

### 3.1. H. pylori Induced TAZ Overexpression and TAZ/TEAD-Dependent Transcriptional Activity

In order to select the most pertinent gastric epithelial cell lines to study TAZ implication in response to *H. pylori* infection, we performed a Western blot to evaluate TAZ expression compared to that of its paralogue YAP in several cell lines commonly used to study *H. pylori*-induced signal transduction in a gastric carcinogenesis context (Figure 1A). The MKN45 cell line showed the highest TAZ/YAP expression ratio among the different cell lines, because only TAZ was expressed and not YAP. The NCI-N87 cell line and the GC07 cell line established from patient-derived GC xenografts showed a high TAZ/YAP ratio and were also selected to perform the following experiments (Figure 1B) [45]. Then, kinetics of TAZ protein expression and TAZ/YAP-mediated TEAD transcriptional activity were performed in these three gastric cell lines in response to infection with the *cag*A-positive 7.13 wild type (WT) *H. pylori* strain (Figure 1C,D). The results showed that *H. pylori* stimulated both TAZ protein expression and TEAD transcriptional activity at 1–2 h post-infection in MKN45 and GC07, and at 2 h post-infection in NCI-N87. This activation was transient and was followed by a return to basal TEAD transcriptional activity at 5 h in all cell lines. After 24 h of infection, TEAD transcriptional activity remained upregulated in MKN45 but repressed in GC07 cells and to a lesser extent in NCI-N87.

A relevant indicator of YAP/TAZ activation is their subcellular localization. Their nuclear localization is associated with their ability to bind nuclear transcription factors such as TEADs to stimulate transcription of target genes. Immunofluorescence staining showed that the basal level of TAZ in cells is weak and localized mostly in the cytoplasm but also in the nucleus of some cells, especially in MKN45 cells (Figure 2A, **arrows**). Upon *H. pylori* infection, we clearly observed a nuclear accumulation of TAZ at 24 h post-infection in the three cell lines, which tended to start as from 2–5 h post-infection in GC07 and NCI-N87 cells (Figure 2A,B). All together, these results show that infection with the 7.13 WT *H. pylori* strain increased TAZ protein expression, nuclear accumulation, and activation of the TAZ–TEAD transcriptional complex activity.

### 3.2. TAZ and ZEB1 Nuclear Co-Overexpression in H. pylori-Infected Gastric Epithelial Cells Was Associated with EMT and Was CagA-Dependent

Kinetic co-culture experiments with *H. pylori* also showed the progressive acquisition of the mesenchymal “hummingbird” phenotype at 5 h and 24 h post-WT *H. pylori* infection in MKN45 and to a lesser extent in NCI-N87 cells, as previously reported [8,13,30] (Figure 2A,C and Appendix A). This mesenchymal phenotype was also observed for the first time in our in-house patient-derived GC07 cell line, in which it was induced as early as 2 h and continued until 24 h post-infection (Figure 2A,C).

To confirm this EMT-like process at the molecular level, we studied the expression and subcellular localization of the EMT marker ZEB1 in parallel to that of TAZ by co-immunofluorescence microscopy over time following 7.13 WT *H. pylori* infection (Figure 2A). A diffuse labelling of ZEB1, mainly in the cytoplasm, was observed in the uninfected basal condition. After 24 h infection, ZEB1 expression was increased and accumulated in the nuclei of the three cell lines, with a concomitant nuclear localization of TAZ, particularly visible in GC07 and MKN45 cells harboring the mesenchymal “hummingbird” phenotype at 5 h and 24 h post-infection (Figure 2A,B). These results show the concomitant TAZ and ZEB1 expression and nuclear localization in cells having undergone EMT upon *H. pylori* infection (Figure 2B).

In order to determine which *H. pylori*-virulence factor was involved in this phenomenon, we performed co-culture experiments of MKN45 cells with the WT or the isogenic mutant of *H. pylori* 7.13 and P12 strains deleted for either *cag*A (∆*cagA*), or a part of the *cag*PAI (∆*cag*PAI), or *cag*E gene, encoding for a scaffolding protein necessary for the T4SS assembly (∆*cagE*) [49] (Appendix A). Both ∆*cag*PAI and ∆*cag*E strains failed to assemble a functional T4SS. The ∆*cag*A strains have a functional T4SS but cannot express the CagA oncoprotein. Western blot analysis of coculture experiments showed that only the WT strains of *H. pylori* were able to increase TAZ protein expression (Appendix A). These results were confirmed by RTqPCR assays showing that only the WT strains were able to increase the expression of CYR61, one of the main TAZ–TEAD target genes, suggesting that this effect could be CagA-dependent. The expression of the EMT markers Vimentin, ZEB1, and Snail, as well as the induction of the “hummingbird” phenotype after 24 h of infection, were also CagA-dependent, as previously reported [8,13] (Appendix A). Altogether, these results show that the *H. pylori*-induced increase in TAZ expression and activity occurred concomitantly to EMT and could be CagA-dependent.

### 3.3. TAZ and ZEB1 Were Co-Upregulated in the Gastric Mucosa of H. pylori-Infected Patients and Mice and in Human Gastric Carcinoma Specimens

To confirm these results in vivo, we evaluated TAZ and ZEB1 expression and subcellular localization by immunohistochemistry on serial tissue sections of gastric mucosa from uninfected and *H. pylori*-infected mice and patients, as well as human gastric carcinoma. Gastric tissue samples from a previously described mouse model of *H. pylori*-induced gastric carcinogenesis were used, corresponding to C57BL/6 mice experimentally infected with the pro-inflammatory *H. pylori* strain HPARE that has a functional *cag*PAI and *cag*A [12,13,30,46]. In mice, as in humans, chronic *H. pylori* infection leads to a sequence of pre-neoplastic lesions of the gastric mucosa evolving with time from chronic gastritis to metaplasia, dysplasia, and gastric carcinoma (gastric intraepithelial neoplasia (GIN) in mice) [5,46]. We previously reported that ZEB1 and other EMT markers were upregulated during this sequence of pre-neoplastic lesions induced by *H. pylori* in mice and humans; they remained upregulated in human gastric carcinoma [13,30,46]. Here, we show that TAZ and ZEB1 are strongly upregulated with a nuclear localization in gastric epithelial cells of *H. pylori*-infected gastric mucosa in mice and humans, when compared to healthy uninfected mucosa (Figure 3A,B,D), thus confirming our in vitro results. Moreover, the nuclear overexpression of TAZ and ZEB1 was detected in poorly differentiated proliferative cells in the isthmus region of *H. pylori*-infected gastric mucosa, which is niche for regenerative epithelial progenitor cells. These stem-like cells are stimulated upon infection to allow tissue regeneration and repair of the epithelial damages induced by chronic *H. pylori* infection and inflammation [13,30,46]. This proliferative region is characterized by a marked Ki67 staining that is more abundant in *H. pylori*-infected conditions compared to uninfected controls.

In human gastric carcinoma, nuclear TAZ and ZEB1 were strongly upregulated compared to healthy gastric mucosa (Figure 3C,D). Notably, the expression and localization of TAZ and ZEB1 were heterogeneous—the center of the tumor masses was less stained than the periphery, where an overexpression of both TAZ and ZEB1 was noted (Figure 3C). An important amount of Ki67-labelled proliferating cells was observed at the tumor periphery and the invasive fronts of the tumors where TAZ and ZEB1 were overexpressed. This suggests an important role for TAZ and ZEB1 in tumor invasion capacities and progression. These ex vivo results confirm those observed in vitro, i.e., that *cag*A-positive *H. pylori* strains induce a co-overexpression of TAZ and ZEB1 in gastric epithelial cells that is associated with a dedifferentiation process; their detection within areas of proliferating cells suggest their implication in both GC initiation and progression.

### 3.4. TAZ Conferred EMT Phenotype and CSC-like Properties via ZEB1 Upregulation in Response to H. pylori Infection

In order to investigate the role of TAZ in *H. pylori*-induced EMT and tumor-initiating properties, we used a silencing RNA strategy (siRNA) to inhibit TAZ. The efficiency of three different siRNA used to inhibit TAZ expression under basal conditions and during *H. pylori* infection was confirmed by Western blot in the three gastric cell lines (Appendix A). The inhibition of TAZ expression in cells transfected with siTAZ-1, siTAZ-2, and siTAZ-3 ranged from 46–73% compared to cells transfected with non-silencing negative control siRNAs (siCtrl). TEAD luciferase reporter assay confirmed these results, showing a strong reduction of TEAD transcriptional activity with the siTAZs even after *H. pylori* infection (from an 80–90% decrease in siTAZ-transfected uninfected conditions to more than 64% in siTAZ-transfected *H. pylori*-infected conditions, compared to siCtrl-transfected uninfected cells; Appendix A). Contrary of MKN45 and GC07, NCI-N87 cells expressed E-cadherin, an essential component of adherent junctions, which acts as a gatekeeper of *H. pylori*-induced EMT to maintain epithelial polarity and integrity [8]. Furthermore, NCI-N87 cells are not invasive in vitro. The consequences of TAZ inhibition on *H. pylori*-induced EMT were therefore evaluated on MKN45 and GC07 cells that are more prone to EMT than NCI-N87 (Figure 4). *H. pylori* induced a significant increase in the expression of several EMT markers such as Vimentin, ZEB1, SNAI1 (except in GC07), and MMP9 in siCtrl-transfected cells, which was abolished or strongly reduced by TAZ silencing in siTAZ-transfected MKN45 and GC07 cells (Figure 4A). To confirm these results, ZEB1 protein expression and subcellular localization was analyzed by immunofluorescence in siRNA-transfected cells upon *H. pylori* infection. Results showed that *H. pylori*-induced ZEB1 overexpression and nuclear accumulation in siCtrl-transfected MKN45 and GC07 cells was no longer observed after TAZ silencing by siTAZ in MKN45 and strongly reduced in GC07 (Figure 4B,C and Appendix A). These results show that TAZ activation controls ZEB1 overexpression and nuclear accumulation in response to *H. pylori* infection.

The invasive properties of cells were then evaluated with invasion assays using type I collagen-coated microporous Transwell inserts (Figure 4D and Appendix A). *H. pylori* infection stimulated invasion in MKN45 cells as previously reported [13], as well as in the GC07 cell line. The silencing of TAZ by the three different siTAZ drastically inhibited *H. pylori*-induced invasive properties in both MKN45 and GC07. Apart from siTAZ1 in MKN45 cells, we observed no effect for the different siTAZ used on basal invasion properties of MKN45 and GC07 cells without *H. pylori* stimulation. 

In the context of *H. pylori*-induced gastric carcinogenesis, EMT has been associated with the acquisition of CSC-related properties by gastric cell lines, such as the capacity to form tumorspheres in vitro [12,13]. To address the role of TAZ in *H. pylori*-induced CSC-like functional properties, we performed a tumorsphere formation assay (Figure 4E,F and Appendix A). The drastic culture conditions of this assay (non-adherent, defined culture media without fetal bovine serum) allows determination of the activity of stem cells, which represent a small subpopulation of cells within the cell lines, which will be the only ones able to survive and self-renew under these conditions. In MKN45, *H. pylori* infection increased the number of tumorspheres as previously reported (Figure 4E,F) [13]. This result was confirmed in NCI-N87 cells and to a lesser extent in GC07 cells (Figure 4F and Appendix A). Remarkably, TAZ inhibition by the three siTAZ showed a full inhibition of the *H. pylori*-induced increase in tumorsphere number in the three cell lines (Figure 4F). There was no effect of siTAZ on the tumorsphere number under basal conditions in MKN45 and NCI-N87 cells, except for siTAZ-3 in NCI-N87. This result indicates that TAZ is not required for self-renewal capacity under basal conditions, but is mandatory in response to *H. pylori* stimulation in order to increase CSC-like tumorigenic properties. 

Altogether, these results highlight the role of TAZ in promoting EMT and the acquisition of invasive and tumorigenic properties related to CSCs in the context of gastric carcinogenesis induced by *H. pylori* infection.

## 4. Discussion

*H. pylori* infection leads to dysregulation of gastric epithelial cell homeostasis by modulating or hijacking several cellular signaling pathways. All these modifications associated with those induced by chronic inflammation and remodeling of the gastric mucosa microenvironment contribute to the initiation and development of gastric carcinogenesis. We previously reported that *cag*A-positive carcinogenic strains of *H. pylori* upregulate the expression of the Hippo tumor suppressor LATS2 and its oncogenic target YAP in a biphasic kinetic manner [30]. Other studies have also reported an upregulation of YAP in response to *H. pylori* [50,51]. Although YAP and TAZ are paralogues, most studies in the field of *H. pylori* and GC have focused on YAP in such a way that the direct implication of TAZ remained unexplored in the context of *H. pylori*-induced gastric carcinogenesis. Results of this study show for the first time that TAZ is upregulated and accumulates in the nuclei of gastric epithelial cells in response to *H. pylori* infection where it binds to its main partners, the TEAD transcription factors. This TAZ nuclear overexpression is associated with the TAZ/TEAD oncogenic transcriptional program illustrated by the increase in *CYR61* expression (Appendix A). CYR61 is an extracellular matrix protein involved in cell adhesion, proliferation, and migration, whose expression has been correlated with poor survival of cardia gastric adenocarcinoma patients; it can be considered a metastatic biomarker for this cancer [52]. This TAZ overexpression and TAZ/TEAD activity seems to occur in a CagA-dependent manner, concomitantly with the ZEB1 overexpression and EMT program. TAZ silencing experiments have shown that TAZ is a major regulator of ZEB1 expression and of the EMT program induced by CagA-positive *H. pylori* strains, and that it controls the induction of invasive and tumorigenic properties. 

We also showed that TAZ is over-expressed in the gastric mucosa of human and mice infected with *H. pylori*. In both cases, TAZ nuclear overexpression was detected in regenerative areas with Ki67-positive epithelial cells. These areas, in the isthmus region of the gastric glands, correspond to an expansion of stem and progenitor cells that are mobilized to ensure epithelium regeneration in response to chronic injury caused by *H. pylori* infection [7,53,54]. Indeed, we previously showed that LATS2 is also overexpressed and nuclear in the same regenerative areas induced by chronic *H. pylori* infection in vivo [30,55].

According to previous study results from our laboratory and others [8,30,53], *H. pylori* induces an increase in EMT markers (ZEB1, Vimentin, Snail, MMP9) and leads to a progressive nuclear translocation and accumulation of ZEB1 protein in the nucleus 5–24 h post *H. pylori* infection in vitro. In vivo, ZEB1 is also overexpressed and nuclear in the stem/progenitor cell regenerative zone of the gastric mucosa in *H. pylori*-infected mice and humans [8,13].

Recent studies concerning EMT state of epithelial cells and their plasticity suggest not only an ON/OFF EMT model, but also a multilayer of EMT features induced by epigenetic events and dysregulation of signaling pathways, called the EMT hybrid state [56,57]. This body of growing evidence of EMT plasticity is well-illustrated in our cellular models of gastric cell lines, with those harboring an epithelial/mesenchymal hybrid state being more prone to a *H. pylori*-induced EMT-like process and acquisition of CSC-like properties. In the context of *H. pylori* infection, E-cadherin expression is sufficient to counteract *H. pylori*-mediated mesenchymal phenotype [58], and on the contrary, downregulation of E-cadherin or other cell–cell junction proteins such as IQ-domain GTPase-activating protein 1 (IQGAP1) predispose cells to overexpress ZEB1 and undergo EMT [12,59]. MKN45 and GC07 PDX-derived cells are poorly cohesive. MKN45 cells have a mutated *cdh1* gene encoding a truncated and non-functional E-cadherin [60]. This may be the most likely cause of their hybrid epithelial/mesenchymal state, which makes them prone to undergoing an EMT-like process more easily than the E-cadherin functional polarized NCI-N87 cells that constrain the EMT-like process induced by *H. pylori*, as we previously reported [8].

Mechanistically, ZEB1 is a transcriptional repressor of epithelial differentiation genes, such as *cdh*1 encoding the E-cadherin adherent junction protein and other cell polarity complex genes, promoting the trans-differentiation characterized by a more or less important loss of epithelial integrity and the acquisition of a highly motile mesenchymal phenotype and invasion properties in cancer [61]. In the gastric carcinogenesis context, experiments on ZEB1 inhibition with siRNA have shown that ZEB1 is essential to the EMT induced in response to *H. pylori* and controls the basal mesenchymal phenotype of some GC cell lines [8,62]. In GC, ZEB1 overexpression is associated with mesenchymal type tumors that have a poor prognosis [62]. In addition to promoting EMT, ZEB1 can also act by repressing the expression of the stemness-inhibiting miR-200b microRNA conferring stemness properties, thereby linking motility and stemness towards a migrating CSC phenotype [8,13,61,63,64]. Its role in the emergence of CSC-like cells with invasive and tumorigenic properties has been demonstrated in several cancers including breast cancer [14] and GC [8,13,62], in which its inhibition by siRNA repressed CSC-like phenotype and properties. Consequently, ZEB1 has been considered as a major actor of cancer cell plasticity in tumor initiation, progression, and metastatic dissemination. A recent study shows that ZEB1, which is mostly known as a transcriptional repressor, could also act as a transcriptional activator by directly interacting with YAP to stimulate the expression of a common set of ZEB1/YAP target genes, including CYR61, which is predictor of metastatic risk and poor survival in aggressive breast cancer [64]. However, Lehmann et al. reported in this study that ZEB1 could directly bind to YAP but not to its paralogue TAZ, due to structural differences between YAP and TAZ—TAZ possesses only one WW domain in the central part of the protein, contrary to YAP that has two WW domains. To our knowledge, the capability of TAZ to bind to ZEB1 has never been investigated in GC. In the study by Lehmann et al., the use of siYAP and siTAZ did not alter ZEB1 expression in aggressive breast cancer cell lines. On the contrary, TAZ inhibition using a lentiviral short hairpin RNA (shRNA) strategy reduced ZEB1 expression in the retinal pigment epithelial cell line (RPE) [65]. We previously showed that these RPE cells are sensitive to *H. pylori* infection and respond, like gastric epithelial cells, through an EMT-like transition involving YAP/TAZ/TEAD activation [30].

One of the remaining questions is how *H. pylori* can activate TAZ, leading to ZEB1 upregulation and induction of an EMT-like process. Different pathways and hypotheses could be associated with TAZ activation in this context. In breast cancer, the protein kinase C (PKC) is a major inducer of the EMT process [66]. *H. pylori*, via the T4SS, activates protein kinase C (PKC), which is responsible for c-Abl phosphorylation leading to its cytoplasmic retention via binding to 14-3-3 protein, promoting EMT-like cell elongation and migration [67]. The c-Abl kinase can phosphorylate TAZ on tyrosine residues, leading to its activation [68]. The link between PKC, c-Abl, and TAZ activation remains to be investigated in the context of *H. pylori*-mediated EMT and gastric carcinogenesis. Other studies have shown that *H. pylori*, via CagA, causes the disruption of the epithelial barrier function by altering the apical junction complex structure by recruiting the tight-junction scaffolding protein Zonula occludens-1 (ZO-1) [7]. ZO-1 and ZO-2 are negative regulators of TAZ by sequestering it at tight junctions and preventing its nuclear translocation and transcriptional activity [69]. We can hypothesize that *H. pylori*, via CagA-induced delocalization of ZO-1 from tight junctions, could release TAZ from the tight junctions and allow its activation and nuclear translocation to participate to EMT. Another pathway linking *H. pylori*-induced signaling to TAZ activation could be the Wnt/β-catenin pathway, activated in response to CagA [59,70]. CagA signaling induces the inhibition of Glycogen synthase kinase 3 beta (GSK-3β) [71] and β-catenin destruction complex, and also the disruption of E-cadherin/β-catenin complexes at cell junctions [72], both leading to β-catenin release and nuclear accumulation, and to a transcriptional program promoting gastric carcinogenesis [73]. Independently of the Hippo pathway, TAZ degradation also depends on phosphorylated β-catenin that bridges TAZ to its ubiquitin ligase Beta-transducin repeat containing E3 ubiquitin protein ligase (β-TrCP). Upon Wnt signaling, activated β-catenin impairs β-TrCP mediated TAZ ubiquitination and degradation, leading to concomitant nuclear accumulation of β-catenin and TAZ [47]. This scenario could contribute to explain the nuclear accumulation of TAZ observed in response to *H. pylori* infection in this study, especially in E-cadherin-deficient gastric cell lines with an epithelial/mesenchymal hybrid state.

In vivo, the nuclear co-overexpression of TAZ and ZEB1 observed in the isthmus regenerative stem and progenitor cell area of the gastric mucosa in *H. pylori*-infected mice and humans is consistent with the TAZ-dependent ZEB1 overexpression and EMT-like process observed in vitro in response to *H. pylori* infection. In vitro, loss-of-function experiments with siTAZ not only resulted in a severe inhibition of EMT marker expression, but also strongly impaired the invasive and tumorigenic properties of gastric cell lines. The nuclear co-overexpression of TAZ and ZEB1 was also detected at the invasive front of the tumors on GC tissue sections, suggesting that TAZ could constitute a marker of invasion and aggressiveness in GC. TAZ expression is a prognostic indicator in colorectal cancer and is required for metastatic activity and chemoresistance of CSC in breast cancer [34,35,38]. In GC, TAZ was reported as a biomarker of aggressiveness as it is associated with the metastatic status of GC patients, and is highly expressed in diffuse type GC associated with a mesenchymal phenotype [40,41]. Furthermore, we recently showed that inhibition of YAP/TAZ interaction with TEADs using Verteporfin inhibits CSC tumorigenic and invasive properties in vitro, as well as tumor growth in GC xenograft models [44], confirming that TAZ, like YAP, could be an interesting biomarker and target in GC.

In conclusion, the present study demonstrated for the first time a novel mechanism by which *H. pylori* upregulates TAZ, which is responsible for the upregulation of ZEB1, one of the most important inducers of EMT and CSC properties in the context of gastric carcinogenesis. Therefore, our findings support the fact that TAZ is a crucial inducer of EMT and invasive and tumorigenic CSC-like properties in response to *H. pylori* infection. The detection of TAZ co-overexpression with ZEB1 and Ki67 in *H. pylori*-induced pre-neoplastic lesions and in invasive GC suggests that TAZ could indeed constitute a biomarker of early transformation in gastric carcinogenesis.

## Figures and Tables

**Figure 1 cells-09-01462-f001:**
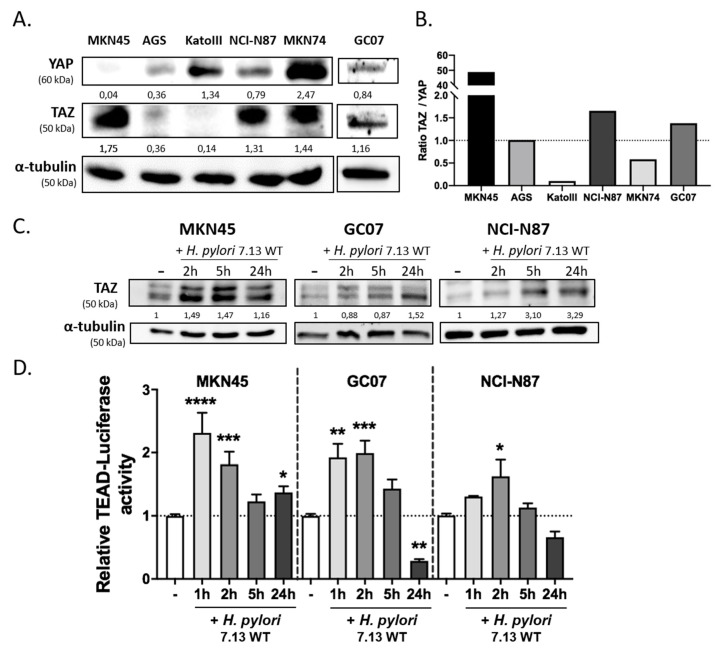
*Helicobacter pylori* increased transcriptional co-activator with PDZ binding motif (TAZ) protein expression and activated TEA domain (TEAD) transcriptional activity. (**A**,**B**) Basal expression of TAZ and yes-associated protein (YAP) evaluated in five different gastric epithelial cell lines and one patient-derived xenograft cell GC07 by Western blot. Numbers under each band correspond to the relative protein expression quantification normalized to the expression of α-tubulin housekeeping protein. (**B**) Representation of the relative expression of TAZ compared to that of YAP in the different gastric epithelial cells. (**C**,**D**) Time-course analyses of co-cultures performed with the 7.13 wild type (WT) *H. pylori* strain and gastric epithelial cell lines MKN45, GC07, and NCI-N87 expressing the most important TAZ/YAP ratio. Uninfected condition corresponds to “–“. (**C**) TAZ protein expression determined during the time-course of 7.13 WT *H. pylori* strain infection by Western blot. Normalization of the quantification was done in comparison to the uninfected condition and α-tubulin housekeeping protein expression. (**D**) Relative TEAD–luciferase activity determined following 7.13 WT *H. pylori* infection. Bars represent means ± standard error of the mean (SEM) of fold changes relative to uninfected cells; *n* = 5, * *p* < 0.05, ** *p* < 0.01, *** *p* < 0.001, **** *p* < 0.0001 vs. uninfected control, Kruskall–Wallis test with Dunn’s post-test.

**Figure 2 cells-09-01462-f002:**
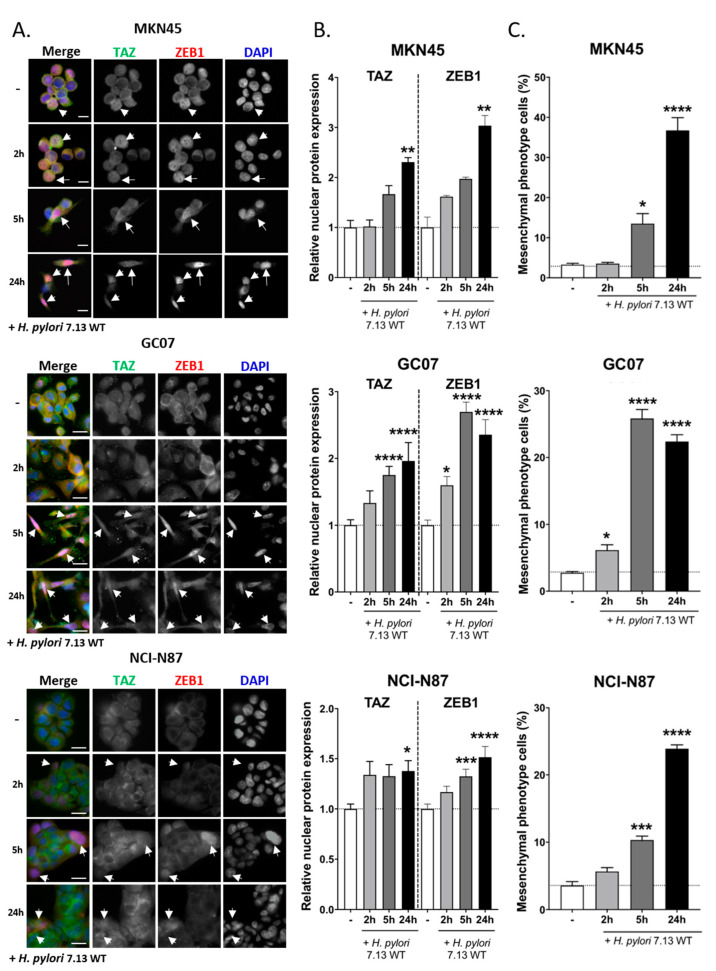
*H. pylori* induced TAZ and ZEB1 co-upregulation associated with their nuclear translocation. (**A**) Expression and subcellular localization of TAZ and ZEB1 assessed by immunofluorescence during the time-course of infection with 7.13 WT *H. pylori* strain. Representative immunofluorescence images of TAZ (in green), ZEB1 (in red), and nuclei stained with DAPI (in blue) in MKN45, GC07, and NCI-N87 cells. Bars scale, 10 µm. White arrows show TAZ and ZEB1 co-accumulation in the nuclei. (**B**) The relative amount of TAZ and ZEB1 nuclear accumulation was quantified by the number of positive- vs. negative-labelled nuclei for MKN45 cells (cells were considered positive when their mean fluorescence intensity was higher than that measured for the uninfected cells plus two standard deviations) and by measuring TAZ and ZEB1 nuclear intensity staining with Fiji software [48] for GC07 and NCI-N87. Bars represent means ± SEM of fold changes relative to uninfected cells. *n* = 3, * *p* < 0.05, ** *p* < 0.01, *** *p* < 0.001, **** *p* < 0.0001 vs. uninfected control in Kruskall–Wallis test with Dunn’s post-test. (**C**) Percentage of cells with mesenchymal “hummingbird” phenotype after *H. pylori* infection in MKN45, GC07, and NCI-N87 cells. Bars represent means ± SEM; *n* = 3, * *p* < 0.05, *** *p* < 0.001, **** *p* < 0.0001 vs. uninfected control in ANOVA test with Dunnett’s post-test.

**Figure 3 cells-09-01462-f003:**
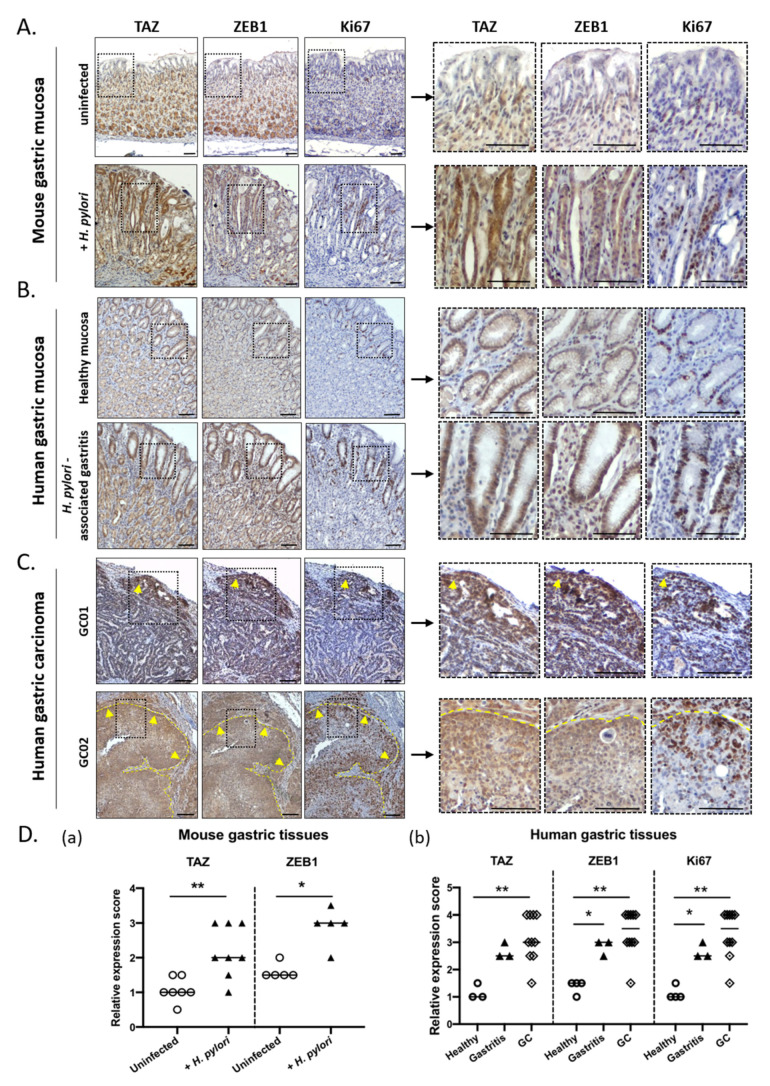
TAZ and ZEB1 were co-upregulated in gastric epithelial cells of *H. pylori*-infected mice and patients and in human gastric carcinoma. Representative images of the immunohistochemistry staining (dark brown) of TAZ, ZEB1, and Ki67 on serial tissue sections. (**A**) Representative images of the gastric mucosa of C57BL/6 mice either uninfected or infected with the *cag*A-positive HPARE *H. pylori* strain for 12 months. (**B**) Representative images of the healthy gastric mucosa of uninfected patients and of patients with *H. pylori*-associated gastritis. (**A**,**B**) The nuclear co-overexpression of TAZ and ZEB1 localized in the regenerative isthmus region characterized by Ki67-positive proliferating cells. (**C**) Representative images of human primary gastric adenocarcinoma (case 01, GC01; case 2, GC02). The yellow dotted line demarcates the limit of the tumor from the stromal microenvironment and the yellow arrows indicate the invasive front of progression of the tumor. Scale bars, 100 µm. (**D**) Scores of the relative quantification of expression of ZEB1, Ki67, and nuclear TAZ (a) in gastric epithelial cells of *H. pylori* HPARE-infected mice (*n* = 6) and uninfected mice (*n* = 6), and of (b) healthy gastric mucosa (*n* = 4) and *H. pylori*-associated gastritis (*n* = 4) patients, and on gastric adenocarcinoma cases (moderately and poorly differentiated, *n* = 10).

**Figure 4 cells-09-01462-f004:**
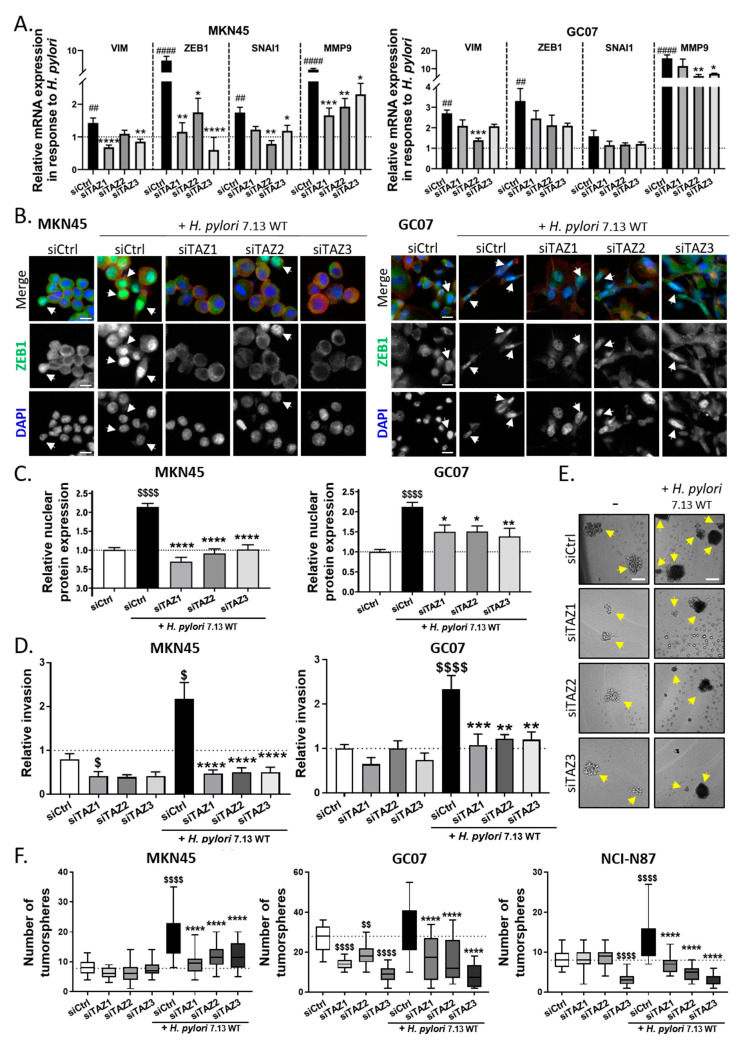
TAZ conferred epithelial–mesenchymal transition (EMT) phenotype and cancer stem cell (CSC)-like properties in response to *H. pylori.* (**A**) Relative mRNA expression of EMT markers Vimentin (VIM), ZEB1, Snail family transcriptional repressor 1 (SNAI1), and Matrix metallopeptidase 9, (MMP9) measured by RT-qPCR in MKN45 and GC07 cells, presented as the ratio of mRNA expression in *H. pylori*-infected vs. uninfected cells for each condition of silencing RNA (siRNA) transfection. Bars represent mean fold changes ± SEM of 4–6 independent experiments relative to uninfected siCtrl. 4 < *n* < 6, * *p* < 0.05, ** *p* < 0.01, *** *p* < 0.001, **** *p* < 0.0001 vs. infected/uninfected siCtrl and # vs. uninfected siCtrl. Kruskall–Wallis test with Dunn’s post-test was used. (**B**) Representative immunofluorescence images of ZEB1 (in green), actin stained with phalloidin (in red), and nuclei stained with DAPI (in blue) in MKN45 and GC07 cells transfected with siCtrl or three different siTAZ and infected with *H. pylori* for 24 h. Scale bars, 10 µm. White arrows show ZEB1 nuclear localization. (**C**) Relative nuclear ZEB1 quantification measured by the nuclear ZEB1 intensity staining. Kruskall–Wallis test with Dunn’s post-test was used. (**D**) After 24 h post *H. pylori* infection and 30 h post siRNA transfection, we seeded 30,000 cells on type I-collagen-coated Transwell inserts; invasive cells were counted 18 h after seeding. Kruskal–Wallis test with Duns post-test was used for MKN45 and ANOVA test with Sidak post-test for GC07. (**E**) Representative images of MKN45 tumorspheres in non-adherent culture conditions. Yellow arrows show tumorsphere formation. Scale bars, 100 µm. (**F**) 30 h post-siTAZ transfection and 24 h post *H. pylori*-infection, we seeded MKN45, GC07, and NCI-N87 cells under non-adherent conditions to investigate their ability to form tumorspheres. Tumorspheres were counted 7 days after cell seeding. ANOVA test with Sidak post-test was used. (**C**,**D**) Bars represent means ± SEM; 3 < *n* < 4, * *p* < 0.05, *** *p* < 0.001, **** *p* < 0.0001 vs. *H. pylori*-infected siCtrl and $ *p* < 0.05, $$ *p* < 0.01, $$$$ *p* < 0.0001 vs. uninfected siCtrl.

**Table 1 cells-09-01462-t001:** RT-qPCR primers sequences used in this study.

Genes	Sequence Forward (5′-3′)	Sequence Reverse (5′-3′)
***CYR61***	ATGAATTGATTGCAGTTGGAAA	TAAAGGGTTGTATAGGATGCGA
***HPRT1***	TGGTCAGGCAGTATAATCCA	GGTCCTTTTCACCAGCAAGCT
***MMP9***	Quantitect primer assay, cat. QT00040040	
***SNAI1***	ACAATGTCTGAAAAGGGACTGTGA	CAGACCAGAGCACCCCATT
***TBP***	GGGCATTATTTGTGCACTGAGA	GCCCAGATAGCAGCACGGT
***VIM***	GGATGCCCTTAAAGGAACCAA	CAACGGCAAAGTTCTCTTCCAT
***ZEB1***	TCCCAACTTATGCCAGGCAC	CAGGAACCACATTTGTCATAGTCAC

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
