# Peer review of "TAZ Controls Helicobacter pylori-Induced Epithelial–Mesenchymal Transition and Cancer Stem Cell-Like Invasive and Tumorigenic Properties"

_cells, 2020, doi:10.3390/cells9061462_

Round 1

Reviewer 1 Report

The manuscript by Tiffon et al present evidence of Cancer Stem cell-like invasive and tumorigenic properties of TAZ and how its induces EMT as a consequence of H.pylori infection. Nevertheless, the manuscript should be modified before final publication. The main comment is the role of H.pylori in a "CagA-dependent manner" which, for this reviewer, is not demonstrated in this article.
Specific comments
Abstract and Graphical abstract should be revised since the role of CagA, as I mentioned before, is not well defined in the results section.
In the introduction, paragraphs related to cagA and phosphorylation should be shortened, particularly phosphorylation since is not part of the methods and results. The final paragraph of the introduction should be also shortened since providing many details of the results is unnecessary in this section.
In methods and results sections, particularly Fig 1, results of MKN-45 cell are contradictory of what is being asked, please explain these results. We also missed a better definition of which H.pylori strain is used in this and following experiments (wt or deleted strains?). In Fig. 2, please explain what is "relative positive nuclei". In Fig. 3 section C, please quantified your IHC results by Q_score or other semiquantitative methods. In results associated with Fig. 4 (line 367), the title of the section should focus on ZEB1 rather TAZ, since it is the direct mechanism for EMT. In Fig. 4, section A, please be consistent in terms of which genes (SNAIL1 and/or MMP9) are display in the images. In section E, an associated image of the spheres is recommended to support these results. Combined experiments of siRNAs, as well as the evaluation of CDH1, should be recommended. However, we understand that this might not be possible in these days.
In Discussion, please be brief about CagA since is not clear from the results the role of cagA in the EMT process. 

Author Response

Comments and Suggestions for Authors

The manuscript by Tiffon et al present evidence of Cancer Stem cell-like invasive and tumorigenic properties of TAZ and how its induces EMT as a consequence of H.pylori infection. Nevertheless, the manuscript should be modified before final publication. The main comment is the role of H.pylori in a "CagA-dependent manner" which, for this reviewer, is not demonstrated in this article.

Specific comments Abstract and Graphical abstract should be revised since the role of CagA, as I mentioned before, is not well defined in the results section. In the introduction, paragraphs related to cagA and phosphorylation should be shortened, particularly phosphorylation since is not part of the methods and results.

- All suggested modifications have been done. We have removed “CagA dependent manner” at several places and/or replaced it by “suggesting that this effect could be CagA-dependent” in the revised version of the manuscript, in the abstract and in the graphical abstract representing the cell signalling (changes in the graphical abstract and lanes: 21, 48, 78 in the annotated revised version of the manuscript with visible corrections).

The final paragraph of the introduction should be also shortened since providing many details of the results is unnecessary in this section.

- The last paragraph of the introduction was shortened as requested. Lines: 104-116 in the annotated revised version of the manuscript with visible corrections.

In methods and results sections, particularly Fig 1, results of MKN-45 cell are contradictory of what is being asked, please explain these results.

- In MKN45 cells, as in GC07 and NCI-N87, we can see an early increase in TAZ expression and TAZ-TEAD transcriptional activity at 1 and 2 h post-infection, though the relative amount of nuclear TAZ at 2 h remained unchanged or lightly tended to increase in GC07 and NCI-N87 but not significantly.  This early activation of the TAZ-TEAD transcriptional activity could be due to an early activation of TAZ already present in the nucleus and/or to the light accumulation of TAZ observed here in GC07 and NCI-N87. In addition, the phosphorylation status of YAP/TAZ is also associated to their activity. We previously showed that in response to H. pylori infection, the ratio of P-YAPSer127 on total YAP (P-YAP/YAP ratio) decreased as from 1 h after infection and was maintained until 2 h post infection (see reference [1], these references are listed at the end of this document). This result suggests that there is an early shift between the inactive phosphorylated form of YAP and the active unphosphorylated form of YAP in response to H. pylori infection. This de-repression of YAP which occurs early in response to H. pylori infection could lead to the early activation of the YAP-TEAD transcriptional activity which was observed [1].  Here, we can hypothesize that the same process occurs with an early increase of the active form of TAZ that could lead to the TAZ-TEAD transcriptional activity observed since 1 h to 2 h post infection (Figure 1D). We could not follow TAZ phosphorylation by western blot or immunofluorescence staining as we did for P-YAP in the study of Molina-Castro [1], we tried but our antibodies directed against P-TAZ were not specific and gave uninterpretable results. A second wave of TAZ activation occurs after 5 h of H. pylori infection, with an abundant TAZ nuclear accumulation in the 3 cell lines (Figure 2A). This was associated with an increase in TAZ-TEAD transcriptional activity at 24 h in MKN45 cells but not in GC07 and NCI-N87 cells, maybe due to a negative retro-control feedback loop as previously reported for YAP [1].  Thus, in response to H. pylori infection, there is a first wave of TAZ activation leading to TAZ-TEAD transcriptional activity since 1 h to 2 h, followed by a significant TAZ nuclear accumulation after 5 h to 24 h.

The paragraph describing the results of the quantification of the nuclear translocation has been rewrote according to these comments. Lines 281-290 in the annotated revised version of the manuscript with visible corrections:

“A relevant indicator of YAP/TAZ activation is their subcellular localisation. Their nuclear localisation is associated with their ability to bind nuclear transcription factors such as TEADs to stimulate transcription of target genes. Immunofluorescence staining showed that the basal level of TAZ in cells is weak and localised mostly in the cytoplasm but also in the nucleus of some cells especially in MKN45 cells (Figure 2A, arrows). Upon H. pylori infection, a nuclear accumulation of TAZ was clearly observed at 24 h post infection in the three cell lines, which tended to start since 2-5 h post-infection in GC07 and NCI-N87 cells (Figure 2A-B). All together, these results show that infection with the 7.13 WT H. pylori strain increases TAZ protein expression, nuclear accumulation and activation of the TAZ-TEAD transcriptional complex activity.”

We also missed a better definition of which H.pylori strain is used in this and following experiments (wt or deleted strains?).

- All experiments were performed with the wild type (WT) 7.13 strain of H. pylori, which encodes a functional type IV secretion system and the CagA oncoprotein. Only some experiments presented in supplementary Figure S1, aiming to investigate cagA and cagPAI roles in H. pylori-mediated effects on TAZ, were performed with the H. pylori 7.13 and P12 WT strains and their isogenic mutants invalidated for cagA and the cagPAI. All other experiments were performed only with the 7.13 WT H. pylori strain. To clarify this point and help the reading and comprehension of the manuscript, we added the name and status of H. pylori strain everywhere in the revised version of the manuscript and on each figure as “H. pylori 7.13 WT” strain (In all figures and everywhere in the revised version of the manuscript).

In Fig. 2, please explain what is "relative positive nuclei".

- The relative amount of TAZ and ZEB1 nuclear accumulation was quantified in the nucleus of each cells by delimiting nuclei using DAPI staining and quantifying the mean fluorescence intensity (pixels shade of grey) of TAZ and ZEB1 labelling with Fiji software [2]. In some experiments, the number of positive versus negative labelled nuclei for MKN45 cells was determined by considering them positive when their mean fluorescence intensity was higher than that of the uninfected cells plus two standard deviations.

According to your question and the unclear legend, we modified the y-axis legend for “Relative nuclear protein expression”, which allows a better comprehension of what is observed.

Lines: 211-2016 and 368-371 in the annotated revised version of the manuscript with visible corrections

In Fig. 3 section C, please quantified your IHC results by Q_score or other semiquantitative methods.

- Relative quantification of the expression of ZEB1, Ki67 and nuclear TAZ in gastric epithelial cells of H. pylori HPARE-infected mice (n=6) and uninfected mice (n=6) and of Healthy gastric mucosa (n=4) and H. pylori-associated gastritis (n=4) patients, and on gastric adenocarcinoma (moderately and poorly differentiated) cases (n= 10) was carried out. Scores were determined in a blind lecture using a scale from 0 to 4 with the following criteria:  0: no staining, 1: <5% of positive epithelial cells; 2: 5 to <25% of positive cells; 3: 25 to <50% of positive cells; 4: ≥50% of positive epithelial cells. This information has been added in the material and methods section of the revised version of the manuscript, as graphs in new Figure 3D, and described in the results section Lines 246-249, 435-438 in the annotated revised version of the manuscript with visible corrections.

In results associated with Fig. 4 (line 367), the title of the section should focus on ZEB1 rather TAZ, since it is the direct mechanism for EMT.

- We modified the title of this section according to your comment: lines 389-390 in the annotated revised version of the manuscript with visible corrections.

 “3.4. TAZ confers EMT phenotype and CSC-like properties via ZEB1 upregulation in response to H. pylori infection.”

In Fig. 4, section A, please be consistent in terms of which genes (SNAIL1 and/or MMP9) are display in the images.

  • According to your comment, we have added RTqPCR results of SNAI1 mRNA expression in GC07 and MMP9 in MKN45 in the revised version of Figure 4A. The variation observed for SNAI1 expression were not significant in GC07, but a trend for an increase of SNAI1 expression induced by H. pylori infection was observed in accordance with the results observed in MKN45. These results are described lines 406 and 483 in the annotated revised version of the manuscript.

In section E, an associated image of the spheres is recommended to support these results.

- As required, representative images of tumourspheres were added to support graphical data in Figure 4E: they are shown in new Figure S2 due to space limitation in Figure 4E.

Combined experiments of siRNAs, as well as the evaluation of CDH1, should be recommended. However, we understand that this might not be possible in these days.

- Instead of combining siRNAs, we tested 3 different siRNAs to confirm the significance of the results obtained. The use of unrelated siRNA sequences targeting the same gene ensured the fact that we are not studying an off-target effect. Concerning the study of CDH1 expression, we and others previously reported that H. pylori induces an increase of the expression of CDH1-encoded E-cadherin protein expression, which may counteracts the H. pylori induced-EMT process to preserve, at least in part, epithelial integrity [3,4]. In addition, CDH1 gene is mutated in MKN45, encoding for a truncated non-functional E-cadherin protein [5].

In Discussion, please be brief about CagA since is not clear from the results the role of cagA in the EMT process. 

- According to this comment, sentences containing “H. pylori CagA dependent manner” were modified to minor these results all along the manuscript and also in discussion section. Lines 474, 475, 605 in the annotated revised version of the manuscript with visible corrections.

- References cited in the responses to the reviewers comments:

  1. Molina-Castro, S.E.; Tiffon, C.; Giraud, J.; Boeuf, H.; Sifre, E.; Giese, A.; Belleannée, G.; Lehours, P.; Bessède, E.; Mégraud, F.; et al. The Hippo Kinase LATS2 Controls Helicobacter pylori-Induced Epithelial-Mesenchymal Transition and Intestinal Metaplasia in Gastric Mucosa. Cell Mol Gastroenterol Hepatol 2020, 9, 257–276, doi:10.1016/j.jcmgh.2019.10.007.
  2. Schindelin, J.; Arganda-Carreras, I.; Frise, E.; Kaynig, V.; Longair, M.; Pietzsch, T.; Preibisch, S.; Rueden, C.; Saalfeld, S.; Schmid, B.; et al. Fiji: an open-source platform for biological-image analysis. Nat. Methods 2012, 9, 676–682, doi:10.1038/nmeth.2019.
  3. Baud, J.; Varon, C.; Chabas, S.; Chambonnier, L.; Darfeuille, F.; Staedel, C. Helicobacter pylori Initiates a Mesenchymal Transition through ZEB1 in Gastric Epithelial Cells. PLoS ONE 2013, 8, e60315, doi:10.1371/journal.pone.0060315.
  4. Bessède, E.; Staedel, C.; Acuña Amador, L.A.; Nguyen, P.H.; Chambonnier, L.; Hatakeyama, M.; Belleannée, G.; Mégraud, F.; Varon, C. Helicobacter pylori generates cells with cancer stem cell properties via epithelial-mesenchymal transition-like changes. Oncogene 2014, 33, 4123–4131, doi:10.1038/onc.2013.380.
  5. Oda, T.; Kanai, Y.; Oyama, T.; Yoshiura, K.; Shimoyama, Y.; Birchmeier, W.; Sugimura, T.; Hirohashi, S. E-cadherin gene mutations in human gastric carcinoma cell lines. Proc. Natl. Acad. Sci. U.S.A. 1994, 91, 1858–1862, doi:10.1073/pnas.91.5.1858.
  6. Oliveira, M.J.; Costa, A.M.; Costa, A.C.; Ferreira, R.M.; Sampaio, P.; Machado, J.C.; Seruca, R.; Mareel, M.; Figueiredo, C. CagA associates with c-Met, E-cadherin, and p120-catenin in a multiproteic complex that suppresses Helicobacter pylori-induced cell-invasive phenotype. J. Infect. Dis. 2009, 200, 745–755, doi:10.1086/604727.
  7. Bessède, E.; Molina, S.; Acuña-Amador, L.; Dubus, P.; Staedel, C.; Chambonnier, L.; Buissonnière, A.; Sifré, E.; Giese, A.; Bénéjat, L.; et al. Deletion of IQGAP1 promotes Helicobacter pylori-induced gastric dysplasia in mice and acquisition of cancer stem cell properties in vitro. Oncotarget 2016, 7, 80688–80699, doi:10.18632/oncotarget.12486.
  8. Wroblewski, L.E.; Piazuelo, M.B.; Chaturvedi, R.; Schumacher, M.; Aihara, E.; Feng, R.; Noto, J.M.; Delgado, A.; Israel, D.A.; Zavros, Y.; et al. Helicobacter pylori targets cancer-associated apical-junctional constituents in gastroids and gastric epithelial cells. Gut 2015, 64, 720–730, doi:10.1136/gutjnl-2014-307650.
  9. Simeone, P.; Trerotola, M.; Franck, J.; Cardon, T.; Marchisio, M.; Fournier, I.; Salzet, M.; Maffia, M.; Vergara, D. The multiverse nature of epithelial to mesenchymal transition. Semin. Cancer Biol. 2019, 58, 1–10, doi:10.1016/j.semcancer.2018.11.004.
  10. Vergara, D.; Simeone, P.; Damato, M.; Maffia, M.; Lanuti, P.; Trerotola, M. The Cancer Microbiota: EMT and Inflammation as Shared Molecular Mechanisms Associated with Plasticity and Progression. J Oncol 2019, 2019, 1253727, doi:10.1155/2019/1253727.
  11. Vergara, D.; Ravaioli, S.; Fonzi, E.; Adamo, L.; Damato, M.; Bravaccini, S.; Pirini, F.; Gaballo, A.; Barbano, R.; Pasculli, B.; et al. Carbonic Anhydrase XII Expression Is Modulated during Epithelial Mesenchymal Transition and Regulated through Protein Kinase C Signaling. Int J Mol Sci 2020, 21, doi:10.3390/ijms21030715.
  12. Posselt, G.; Wiesauer, M.; Chichirau, B.E.; Engler, D.; Krisch, L.M.; Gadermaier, G.; Briza, P.; Schneider, S.; Boccellato, F.; Meyer, T.F.; et al. Helicobacter pylori-controlled c-Abl localization promotes cell migration and limits apoptosis. Cell Commun. Signal 2019, 17, 10, doi:10.1186/s12964-019-0323-9.
  13. Jang, E.J.; Jeong, H.; Han, K.H.; Kwon, H.M.; Hong, J.-H.; Hwang, E.S. TAZ suppresses NFAT5 activity through tyrosine phosphorylation. Mol. Cell. Biol. 2012, 32, 4925–4932, doi:10.1128/MCB.00392-12.
  14. Amieva, M.R.; Vogelmann, R.; Covacci, A.; Tompkins, L.S.; Nelson, W.J.; Falkow, S. Disruption of the epithelial apical-junctional complex by Helicobacter pylori CagA. Science 2003, 300, 1430–1434, doi:10.1126/science.1081919.
  15. Remue, E.; Meerschaert, K.; Oka, T.; Boucherie, C.; Vandekerckhove, J.; Sudol, M.; Gettemans, J. TAZ interacts with zonula occludens-1 and -2 proteins in a PDZ-1 dependent manner. FEBS Lett. 2010, 584, 4175–4180, doi:10.1016/j.febslet.2010.09.020.
  16. Nagy, T.A.; Frey, M.R.; Yan, F.; Israel, D.A.; Polk, D.B.; Peek, R.M. Helicobacter pylori regulates cellular migration and apoptosis by activation of phosphatidylinositol 3-kinase signaling. J. Infect. Dis. 2009, 199, 641–651, doi:10.1086/596660.
  17. Nagy, T.A.; Wroblewski, L.E.; Wang, D.; Piazuelo, M.B.; Delgado, A.; Romero-Gallo, J.; Noto, J.; Israel, D.A.; Ogden, S.R.; Correa, P.; et al. β-Catenin and p120 mediate PPARδ-dependent proliferation induced by Helicobacter pylori in human and rodent epithelia. Gastroenterology 2011, 141, 553–564, doi:10.1053/j.gastro.2011.05.004.
  18. Murata-Kamiya, N.; Kurashima, Y.; Teishikata, Y.; Yamahashi, Y.; Saito, Y.; Higashi, H.; Aburatani, H.; Akiyama, T.; Peek, R.M.; Azuma, T.; et al. Helicobacter pylori CagA interacts with E-cadherin and deregulates the beta-catenin signal that promotes intestinal transdifferentiation in gastric epithelial cells. Oncogene 2007, 26, 4617–4626, doi:10.1038/sj.onc.1210251.
  19. Franco, A.T.; Israel, D.A.; Washington, M.K.; Krishna, U.; Fox, J.G.; Rogers, A.B.; Neish, A.S.; Collier-Hyams, L.; Perez-Perez, G.I.; Hatakeyama, M.; et al. Activation of beta-catenin by carcinogenic Helicobacter pylori. Proc. Natl. Acad. Sci. U.S.A. 2005, 102, 10646–10651, doi:10.1073/pnas.0504927102.
  20. Azzolin, L.; Panciera, T.; Soligo, S.; Enzo, E.; Bicciato, S.; Dupont, S.; Bresolin, S.; Frasson, C.; Basso, G.; Guzzardo, V.; et al. YAP/TAZ incorporation in the β-catenin destruction complex orchestrates the Wnt response. Cell 2014, 158, 157–170, doi:10.1016/j.cell.2014.06.013.
  21. Ohnishi, N.; Yuasa, H.; Tanaka, S.; Sawa, H.; Miura, M.; Matsui, A.; Higashi, H.; Musashi, M.; Iwabuchi, K.; Suzuki, M.; et al. Transgenic expression of Helicobacter pylori CagA induces gastrointestinal and hematopoietic neoplasms in mouse. Proc. Natl. Acad. Sci. U.S.A. 2008, 105, 1003–1008, doi:10.1073/pnas.0711183105.
  22. Neal, J.T.; Peterson, T.S.; Kent, M.L.; Guillemin, K. H. pylori virulence factor CagA increases intestinal cell proliferation by Wnt pathway activation in a transgenic zebrafish model. Dis Model Mech 2013, 6, 802–810, doi:10.1242/dmm.011163.
  23. Varon, C.; Dubus, P.; Mazurier, F.; Asencio, C.; Chambonnier, L.; Ferrand, J.; Giese, A.; Senant-Dugot, N.; Carlotti, M.; Mégraud, F. Helicobacter pylori infection recruits bone marrow-derived cells that participate in gastric preneoplasia in mice. Gastroenterology 2012, 142, 281–291, doi:10.1053/j.gastro.2011.10.036.
  24. Belair, C.; Baud, J.; Chabas, S.; Sharma, C.M.; Vogel, J.; Staedel, C.; Darfeuille, F. Helicobacter pylori interferes with an embryonic stem cell micro RNA cluster to block cell cycle progression. Silence 2011, 2, 7, doi:10.1186/1758-907X-2-7.

Reviewer 2 Report

In this manuscript, the authors investigated the role of TAZ in the early steps of H. pylori-mediated gastric carcinogenesis. In cell models, H. pylori infection induces TAZ protein expression and nuclear accumulation. In vivo results confirm that cagA-positive H. pylori strains induce a co-overexpression of TAZ and ZEB1 in gastric epithelial cells. By siRNA, the authors demonstrated that cagA-induced EMT is TAZ mediated. Overall, this is a well interesting study with an excellent research design.

Authors investigated the role of cagA-induced EMT in cells with basal expression of mesenchymal markers (hybrid E/M models ?). Therefore, I wonder how cagA can induce EMT by TAZ in cells with Epithelial features. I suggest clarifying the E/M status of cell models used in this work. The difference is substantial. CagA can induce a transition in a hybrid model and a complete EMT activation in a cell with only E features. In the first case, the modulation of some EMT factors or protein kinase (i.e. the PKA/PKC balance in MCF-10A) is sufficient, in the second case, a more robust (i.e. epigenetic changes) effect is mandatory.

Recently, we discussed these topics in these two papers: - Semin Cancer Biol
. 2019 Oct;58:1-10. doi: 10.1016/j.semcancer.2018.11.004.  / J Oncol
. 2019 Oct 20;2019:1253727. doi: 10.1155/2019/1253727.

I hope the authors may found these useful for their discussion.

Author Response

Dear Editor and reviewers,

Please find below our point-by-point responses to the reviewers comments. All points raised by the reviewers have been addressed, and new data and paragraphs have been added as requested.

I hope you will consider positively our revised version of the manuscript for publication in Cells.

Best regards

Christine Varon, corresponding author

Reviewer 2

Open Review

English language and style

( ) Extensive editing of English language and style required
( ) Moderate English changes required
(x) English language and style are fine/minor spell check required
( ) I don't feel qualified to judge about the English language and style

Yes

Can be improved

Must be improved

Not applicable

Does the introduction provide sufficient background and include all relevant references?

(x)

( )

( )

( )

Is the research design appropriate?

(x)

( )

( )

( )

Are the methods adequately described?

(x)

( )

( )

( )

Are the results clearly presented?

(x)

( )

( )

( )

Are the conclusions supported by the results?

( )

(x)

( )

( )

Comments and Suggestions for Authors

In this manuscript, the authors investigated the role of TAZ in the early steps of H. pylori-mediated gastric carcinogenesis. In cell models, H. pylori infection induces TAZ protein expression and nuclear accumulation. In vivo results confirm that cagA-positive H. pylori strains induce a co-overexpression of TAZ and ZEB1 in gastric epithelial cells. By siRNA, the authors demonstrated that cagA-induced EMT is TAZ mediated. Overall, this is a well interesting study with an excellent research design.

Authors investigated the role of cagA-induced EMT in cells with basal expression of mesenchymal markers (hybrid E/M models ?). Therefore, I wonder how cagA can induce EMT by TAZ in cells with Epithelial features. I suggest clarifying the E/M status of cell models used in this work. The difference is substantial. CagA can induce a transition in a hybrid model and a complete EMT activation in a cell with only E features. In the first case, the modulation of some EMT factors or protein kinase (i.e. the PKA/PKC balance in MCF-10A) is sufficient, in the second case, a more robust (i.e. epigenetic changes) effect is mandatory.

Recently, we discussed these topics in these two papers: - Semin Cancer Biol
. 2019 Oct;58:1-10. doi: 10.1016/j.semcancer.2018.11.004.  / J Oncol
. 2019 Oct 20;2019:1253727. doi: 10.1155/2019/1253727.

I hope the authors may found these useful for their discussion.

- Thank you for your relevant comments regarding the epithelial/mesenchymal hybrid state of our cell lines. E-cadherin plays a major role is the epithelial phenotype maintenance. In H. pylori infection context, E-cadherin expression is sufficient to counteract H. pylori-mediated mesenchymal phenotype [6] and on the contrary, downregulation of E-cadherin or other structural components of the cell-cell junctions such as IQGAP1 can predispose cells to undergo an EMT-like process in response to H. pylori [7,8].  MKN45 cells have a mutated CDH1 gene encoding a truncated and non-functional form of E-cadherin, which may be the most likely cause of their hybrid E/M state predisposing them to undergo EMT rather than a total reprogramming with genetic and epigenetic modification. Concerning our GC07 PDX-derived cell line, it is probably the same hybrid E/M state because of their poor cohesive phenotype probably due to the downregulation and/or inactivation of E-cadherin or other cell-cell junction protein. For the NCI-N87 cells, they have a functional E-cadherin protein associated with a cohesive function and a polarized phenotype; their epithelial phenotype constrains the EMT-like process in response to H. pylori infection as we previously reported [3].

Your comments were relevant and helpful to shed light on this intermediate E/M state which can be useful to understand the carcinogenesis process induced by H. pylori and for further developing therapeutic perspective based on the wide range of epithelial/mesenchymal state of tumours. This reflexion also stimulated us to explore and discuss the putative upstream regulatory mechanisms linking H. pylori infection and CagA to TAZ activation and accumulation.

We have discussed all of this and added the suggested reviews as well as additional references in the discussion section of the revised version of the manuscript, lines 514 to 528 and 558 to 584 in the annotated revised version of the manuscript with visible corrections.

Lines 514 – 528: “Recent studies concerning EMT state of epithelial cells and their plasticity, suggest not only an ON/OFF EMT model, but also a multilayer of EMT features induced by epigenetic events and dysregulation of signalling pathway, called EMT hybrid state [9,10]. This body growing of evidence of EMT plasticity is well-illustrated in our cellular models of gastric cell lines, those harbouring and epithelial/mesenchymal hybrid state being more prone to H. pylori-induced EMT-like process and acquisition of CSC-like properties. In H. pylori infection context, E-cadherin expression is sufficient to counteracts H. pylori-mediated mesenchymal phenotype [6] and on the contrary, downregulation of E-cadherin or other cell-cell junctions proteins such as IQGAP1 predispose cells to overexpress ZEB1 and undergo EMT [7,8]. MKN45 and GC07 PDX-derived cells are poorly cohesive. MKN45 cells have a mutated CDH1 gene encoding a truncated and non-functional E-cadherin [5]. This may be the most likely cause of their hybrid epithelial/mesenchymal state, which prone them to undergo more easily an EMT-like process than the E-cadherin functional polarized NCI-N87 cells that constrains the EMT-like process induced by H. pylori as we previously reported [3]. “

 Lines 558-584: “One of the remaining questions is how H. pylori can activate TAZ leading to ZEB1 upregulation and induction an EMT-like process? Different pathways and hypothesizes could be associated to TAZ activation in this context. In breast cancer, the Protein Kinase C (PKC) is a major inducer of EMT process [11]. H. pylori, via the T4SS, activates the Protein Kinase C (PKC), which is responsible for c-Abl phosphorylation leading to its cytoplasmic retention via binding to 14-3-3 protein, promoting EMT-like cell elongation and migration [12]. The c-Abl kinase can phosphorylates TAZ on tyrosine residues, leading to its activation [13]. The link between PKC, c-Abl and TAZ activation remains to be investigated in the context of H. pylori-mediated EMT and gastric carcinogenesis. Other studies have shown that H. pylori, via CagA, causes the disruption of the epithelial barrier function by altering the apical junction complexes structure by recruiting the tight-junction scaffolding protein ZO-1 [14]. ZO-1 and ZO-2 are negative regulator of TAZ by sequestering it at tight junctions and preventing its nuclear translocation and transcriptional activity [15]. We can hypothesize that H. pylori, via CagA-induced delocalization of ZO-1 from tight junctions, could release TAZ from the tight junctions and allow its activation and nuclear translocation to participate to EMT. Another pathway linking H. pylori-induced signalling to TAZ activation could be the Wnt/β-catenin pathway activated in response to CagA [8,16]. CagA signalling induces the inhibition of GSK-3β [17] and β-catenin destruction complex, and also the disruption of E-cadherin/β-catenin complexes at cell junctions [18], both leading to β-catenin release and nuclear accumulation, and to a transcriptional programme promoting gastric carcinogenesis [19]. Independently of the Hippo pathway, TAZ degradation also depends on phosphorylated β-catenin that bridges TAZ to its ubiquitin ligase β-TrCP. Upon Wnt signalling, activated β-catenin impairs β-TrCP mediated TAZ ubiquitination and degradation and leads to concomitant nuclear accumulation of β-catenin and TAZ [20]. This scenario could contribute to explain the nuclear accumulation of TAZ observed in response to H. pylori infection in this study, especially in E-cadherin-deficient gastric cell lines with an epithelial/mesenchymal hybrid state.”

- References cited in the responses to the reviewers comments:

  1. Molina-Castro, S.E.; Tiffon, C.; Giraud, J.; Boeuf, H.; Sifre, E.; Giese, A.; Belleannée, G.; Lehours, P.; Bessède, E.; Mégraud, F.; et al. The Hippo Kinase LATS2 Controls Helicobacter pylori-Induced Epithelial-Mesenchymal Transition and Intestinal Metaplasia in Gastric Mucosa. Cell Mol Gastroenterol Hepatol 2020, 9, 257–276, doi:10.1016/j.jcmgh.2019.10.007.
  2. Schindelin, J.; Arganda-Carreras, I.; Frise, E.; Kaynig, V.; Longair, M.; Pietzsch, T.; Preibisch, S.; Rueden, C.; Saalfeld, S.; Schmid, B.; et al. Fiji: an open-source platform for biological-image analysis. Nat. Methods 2012, 9, 676–682, doi:10.1038/nmeth.2019.
  3. Baud, J.; Varon, C.; Chabas, S.; Chambonnier, L.; Darfeuille, F.; Staedel, C. Helicobacter pylori Initiates a Mesenchymal Transition through ZEB1 in Gastric Epithelial Cells. PLoS ONE 2013, 8, e60315, doi:10.1371/journal.pone.0060315.
  4. Bessède, E.; Staedel, C.; Acuña Amador, L.A.; Nguyen, P.H.; Chambonnier, L.; Hatakeyama, M.; Belleannée, G.; Mégraud, F.; Varon, C. Helicobacter pylori generates cells with cancer stem cell properties via epithelial-mesenchymal transition-like changes. Oncogene 2014, 33, 4123–4131, doi:10.1038/onc.2013.380.
  5. Oda, T.; Kanai, Y.; Oyama, T.; Yoshiura, K.; Shimoyama, Y.; Birchmeier, W.; Sugimura, T.; Hirohashi, S. E-cadherin gene mutations in human gastric carcinoma cell lines. Proc. Natl. Acad. Sci. U.S.A. 1994, 91, 1858–1862, doi:10.1073/pnas.91.5.1858.
  6. Oliveira, M.J.; Costa, A.M.; Costa, A.C.; Ferreira, R.M.; Sampaio, P.; Machado, J.C.; Seruca, R.; Mareel, M.; Figueiredo, C. CagA associates with c-Met, E-cadherin, and p120-catenin in a multiproteic complex that suppresses Helicobacter pylori-induced cell-invasive phenotype. J. Infect. Dis. 2009, 200, 745–755, doi:10.1086/604727.
  7. Bessède, E.; Molina, S.; Acuña-Amador, L.; Dubus, P.; Staedel, C.; Chambonnier, L.; Buissonnière, A.; Sifré, E.; Giese, A.; Bénéjat, L.; et al. Deletion of IQGAP1 promotes Helicobacter pylori-induced gastric dysplasia in mice and acquisition of cancer stem cell properties in vitro. Oncotarget 2016, 7, 80688–80699, doi:10.18632/oncotarget.12486.
  8. Wroblewski, L.E.; Piazuelo, M.B.; Chaturvedi, R.; Schumacher, M.; Aihara, E.; Feng, R.; Noto, J.M.; Delgado, A.; Israel, D.A.; Zavros, Y.; et al. Helicobacter pylori targets cancer-associated apical-junctional constituents in gastroids and gastric epithelial cells. Gut 2015, 64, 720–730, doi:10.1136/gutjnl-2014-307650.
  9. Simeone, P.; Trerotola, M.; Franck, J.; Cardon, T.; Marchisio, M.; Fournier, I.; Salzet, M.; Maffia, M.; Vergara, D. The multiverse nature of epithelial to mesenchymal transition. Semin. Cancer Biol. 2019, 58, 1–10, doi:10.1016/j.semcancer.2018.11.004.
  10. Vergara, D.; Simeone, P.; Damato, M.; Maffia, M.; Lanuti, P.; Trerotola, M. The Cancer Microbiota: EMT and Inflammation as Shared Molecular Mechanisms Associated with Plasticity and Progression. J Oncol 2019, 2019, 1253727, doi:10.1155/2019/1253727.
  11. Vergara, D.; Ravaioli, S.; Fonzi, E.; Adamo, L.; Damato, M.; Bravaccini, S.; Pirini, F.; Gaballo, A.; Barbano, R.; Pasculli, B.; et al. Carbonic Anhydrase XII Expression Is Modulated during Epithelial Mesenchymal Transition and Regulated through Protein Kinase C Signaling. Int J Mol Sci 2020, 21, doi:10.3390/ijms21030715.
  12. Posselt, G.; Wiesauer, M.; Chichirau, B.E.; Engler, D.; Krisch, L.M.; Gadermaier, G.; Briza, P.; Schneider, S.; Boccellato, F.; Meyer, T.F.; et al. Helicobacter pylori-controlled c-Abl localization promotes cell migration and limits apoptosis. Cell Commun. Signal 2019, 17, 10, doi:10.1186/s12964-019-0323-9.
  13. Jang, E.J.; Jeong, H.; Han, K.H.; Kwon, H.M.; Hong, J.-H.; Hwang, E.S. TAZ suppresses NFAT5 activity through tyrosine phosphorylation. Mol. Cell. Biol. 2012, 32, 4925–4932, doi:10.1128/MCB.00392-12.
  14. Amieva, M.R.; Vogelmann, R.; Covacci, A.; Tompkins, L.S.; Nelson, W.J.; Falkow, S. Disruption of the epithelial apical-junctional complex by Helicobacter pylori CagA. Science 2003, 300, 1430–1434, doi:10.1126/science.1081919.
  15. Remue, E.; Meerschaert, K.; Oka, T.; Boucherie, C.; Vandekerckhove, J.; Sudol, M.; Gettemans, J. TAZ interacts with zonula occludens-1 and -2 proteins in a PDZ-1 dependent manner. FEBS Lett. 2010, 584, 4175–4180, doi:10.1016/j.febslet.2010.09.020.
  16. Nagy, T.A.; Frey, M.R.; Yan, F.; Israel, D.A.; Polk, D.B.; Peek, R.M. Helicobacter pylori regulates cellular migration and apoptosis by activation of phosphatidylinositol 3-kinase signaling. J. Infect. Dis. 2009, 199, 641–651, doi:10.1086/596660.
  17. Nagy, T.A.; Wroblewski, L.E.; Wang, D.; Piazuelo, M.B.; Delgado, A.; Romero-Gallo, J.; Noto, J.; Israel, D.A.; Ogden, S.R.; Correa, P.; et al. β-Catenin and p120 mediate PPARδ-dependent proliferation induced by Helicobacter pylori in human and rodent epithelia. Gastroenterology 2011, 141, 553–564, doi:10.1053/j.gastro.2011.05.004.
  18. Murata-Kamiya, N.; Kurashima, Y.; Teishikata, Y.; Yamahashi, Y.; Saito, Y.; Higashi, H.; Aburatani, H.; Akiyama, T.; Peek, R.M.; Azuma, T.; et al. Helicobacter pylori CagA interacts with E-cadherin and deregulates the beta-catenin signal that promotes intestinal transdifferentiation in gastric epithelial cells. Oncogene 2007, 26, 4617–4626, doi:10.1038/sj.onc.1210251.
  19. Franco, A.T.; Israel, D.A.; Washington, M.K.; Krishna, U.; Fox, J.G.; Rogers, A.B.; Neish, A.S.; Collier-Hyams, L.; Perez-Perez, G.I.; Hatakeyama, M.; et al. Activation of beta-catenin by carcinogenic Helicobacter pylori. Proc. Natl. Acad. Sci. U.S.A. 2005, 102, 10646–10651, doi:10.1073/pnas.0504927102.
  20. Azzolin, L.; Panciera, T.; Soligo, S.; Enzo, E.; Bicciato, S.; Dupont, S.; Bresolin, S.; Frasson, C.; Basso, G.; Guzzardo, V.; et al. YAP/TAZ incorporation in the β-catenin destruction complex orchestrates the Wnt response. Cell 2014, 158, 157–170, doi:10.1016/j.cell.2014.06.013.
  21. Ohnishi, N.; Yuasa, H.; Tanaka, S.; Sawa, H.; Miura, M.; Matsui, A.; Higashi, H.; Musashi, M.; Iwabuchi, K.; Suzuki, M.; et al. Transgenic expression of Helicobacter pylori CagA induces gastrointestinal and hematopoietic neoplasms in mouse. Proc. Natl. Acad. Sci. U.S.A. 2008, 105, 1003–1008, doi:10.1073/pnas.0711183105.
  22. Neal, J.T.; Peterson, T.S.; Kent, M.L.; Guillemin, K. H. pylori virulence factor CagA increases intestinal cell proliferation by Wnt pathway activation in a transgenic zebrafish model. Dis Model Mech 2013, 6, 802–810, doi:10.1242/dmm.011163.
  23. Varon, C.; Dubus, P.; Mazurier, F.; Asencio, C.; Chambonnier, L.; Ferrand, J.; Giese, A.; Senant-Dugot, N.; Carlotti, M.; Mégraud, F. Helicobacter pylori infection recruits bone marrow-derived cells that participate in gastric preneoplasia in mice. Gastroenterology 2012, 142, 281–291, doi:10.1053/j.gastro.2011.10.036.
  24. Belair, C.; Baud, J.; Chabas, S.; Sharma, C.M.; Vogel, J.; Staedel, C.; Darfeuille, F. Helicobacter pylori interferes with an embryonic stem cell micro RNA cluster to block cell cycle progression. Silence 2011, 2, 7, doi:10.1186/1758-907X-2-7.

Reviewer 3 Report

Dear Editor,

The manuscript entitled “TAZ Controls Helicobacter pylori-Induced Epithelial-2 Mesenchymal Transition and Cancer Stem Cell-Like 3 Invasive and Tumourigenic Properties” describes that the role of TAZ in the early steps of H. pylori-mediated gastric carcinogenesis. TAZ implication in EMT, invasion, and CSC-related tumourigenic properties are evaluated in three gastric epithelial cell lines infected by H. pylori. Authors show that H. pylori infection increases TAZ nuclear expression and TEADs transcriptional activity in a CagA-dependent manner. Nuclear TAZ and ZEB1 are co-overexpressed in cells harbouring a mesenchymal phenotype in vitro, and in areas of regenerative hyperplasia in gastric mucosa of H. pylori-infected patients and experimentally-infected mice, as well as at the invasive front of gastric carcinoma. TAZ silencing reduces ZEB1 expression, EMT phenotype and strongly inhibits invasion and tumoursphere formation induced by H. pylori. In conclusion, TAZ activation in response to H. pylori infection contributes to H. pylori-induced EMT, invasion, and CSC-like tumourigenic properties. TAZ overexpression in H. pylori-induced pre-neoplastic lesions and in GC could therefore constitute a biomarker of early transformation in gastric carcinogenesis.

The followings need to be addressed.

  1. Authors should use animal experiments to identify the H. pylori (wild type and mutant)-mediated tumourigenic properties instead of in vitro cell sphere formation.
  2. What effects dose H. pylori (wild type and mutant) contribute to cell proliferation and growth in vitro experiments?
  3. Authors should show the image of cell sphere formation induced H. pylori (wild type and mutant), and measure the markers of sphere formation.
  4. Authors should measure the nuclear TAZ and ZEB1 in H. pylori (wild type and mutant)-infected cells, using immunoblotting assay. IF in this study is difficult to be observed.
  5. Authors should show the image of invasion induced H. pylori (wild type and mutant).
  6. Authors should measure the protein level of EMT markers in this study. Protein has the real function in regulating cell physiology, responses, behaviors et al.

Author Response

Dear Editor and reviewers,

Please find below our point-by-point responses to the reviewers comments. All points raised by the reviewers have been addressed, and new data and paragraphs have been added as requested.

I hope you will consider positively our revised version of the manuscript for publication in Cells.

Best regards

Christine Varon, corresponding author

Reviewer 3

Comments and Suggestions for Authors

English language and style

( ) Extensive editing of English language and style required
( ) Moderate English changes required
( ) English language and style are fine/minor spell check required
(x) I don't feel qualified to judge about the English language and style

Yes

Can be improved

Must be improved

Not applicable

Does the introduction provide sufficient background and include all relevant references?

( )

(x)

( )

( )

Is the research design appropriate?

( )

(x)

( )

( )

Are the methods adequately described?

( )

(x)

( )

( )

Are the results clearly presented?

( )

(x)

( )

( )

Are the conclusions supported by the results?

( )

(x)

( )

( )

Dear Editor,

The manuscript entitled “TAZ Controls Helicobacter pylori-Induced Epithelial-2 Mesenchymal Transition and Cancer Stem Cell-Like 3 Invasive and Tumourigenic Properties” describes that the role of TAZ in the early steps of H. pylori-mediated gastric carcinogenesis. TAZ implication in EMT, invasion, and CSC-related tumourigenic properties are evaluated in three gastric epithelial cell lines infected by H. pylori. Authors show that H. pylori infection increases TAZ nuclear expression and TEADs transcriptional activity in a CagA-dependent manner. Nuclear TAZ and ZEB1 are co-overexpressed in cells harbouring a mesenchymal phenotype in vitro, and in areas of regenerative hyperplasia in gastric mucosa of H. pylori-infected patients and experimentally-infected mice, as well as at the invasive front of gastric carcinoma. TAZ silencing reduces ZEB1 expression, EMT phenotype and strongly inhibits invasion and tumoursphere formation induced by H. pylori. In conclusion, TAZ activation in response to H. pylori infection contributes to H. pylori-induced EMT, invasion, and CSC-like tumourigenic properties. TAZ overexpression in H. pylori-induced pre-neoplastic lesions and in GC could therefore constitute a biomarker of early transformation in gastric carcinogenesis.

The followings need to be addressed.

  1. Authors should use animal experiments to identify the H. pylori (wild type and mutant)-mediated tumourigenic properties instead of in vitro cell sphere formation.

- H. pylori-mediated tumourigenic properties have already been demonstrated in animal studies and confirm the implication of CagA oncoprotein in gastric carcinogenesis. The team of Masanori Hatakeyama was the first to generate transgenic mice expressing CagA in an ubiquitous and constitutive manner throughout the body or especially in the stomach thanks to a tissue-specific promoter. In this mouse model, expression of CagA initiated malignancies in the body (lymphoma) and more importantly gastric carcinoma, providing a direct evidence for the role of CagA as the first bacterial oncoprotein [21]. Expression of CagA in transgenic zebrafish induced intestinal neoplasia after p53 induced mutation [22]. Franco et al have also shown that CagA-positive strains of H. pylori, and especially the 7.13 WT strain used in our study, induces gastric carcinogenesis in mongolian gerbil model, an animal model which is more susceptible to H. pylori colonization than mouse models [19]. Previous works from our laboratory have confirmed these results by using cagA-positive H. pylori strains (among which the strain HPARE used in mice infection experiments in this study) to induce gastric carcinogenesis lesions in WT and transgenic mouse model [1,4,7,23].

  1. What effects dose H. pylori (wild type and mutant) contribute to cell proliferation and growth in vitro experiments?

- Previous works from our laboratory and others have found that H. pylori infection leads to a decrease of cell proliferation in several gastric epithelial cell lines. Previous works from our laboratory have shown that cagA+ H. pylori strains induce a cell-cycle arrest in AGS cells at the G1/S transition [1,24]. Indeed, the growth rate of H. pylori 7.13 WT-infected MKN45 cells at 24 h of infection assessed by cell numeration was reduced by 37,98% +/- 3,04% compared to uninfected cells. Similar growth inhibition was observed in H. pylori infected NCI-N87 cells (42,44% +/- 12,34% at 48 h post infection compared to uninfected cells) and in our PDX-derived GC07 cells (20,01 +/- 16,98 at 24 h post infection compared to uninfected condition).

  1. Authors should show the image of cell sphere formation induced H. pylori (wild type and mutant), and measure the markers of sphere formation.

- According to your suggestion, we have added representative images of sphere formation observed upon siRNA transfection and H. pylori infection (new Figure 4E, new Figure S2E and corresponding figure legends). The determination of the number of spheres formed in non-adherent and serum-free culture conditions reflects the number of CSCs in the cell population[4,7]. In this study, we have shown that H. pylori is able to increase the number of spheres formed by infected cells and so acts on the CSC population. The inhibition of TAZ expression decreased the number of CSCs in the infected cell population, reflected by the decrease in the number of spheres formed. Experiments were performed only with the 7.13 WT H. pylori strain. Previous experiments from our team with cagA and cagPAI mutant strains of H. pylori have shown that H. pylori-induced increase in sphere formation is CagA-dependent [4].

  1. Authors should measure the nuclear TAZ and ZEB1 in H. pylori (wild type and mutant)-infected cells, using immunoblotting assay. IF in this study is difficult to be observed.

- We tried to measure the nuclear TAZ and ZEB1 expression by performing western blot assay, but our nuclear fractionation protocol did not correctly separate the cytoplasmic and nuclear fractions; in addition, our anti-ZEB1 antibody (from Bethyl) does not work for western blot in our cell lines (works only for immunofluorescence and immunohistochemistry). We choose to measure the nuclear TAZ and ZEB1 expression by performing co-immunofluorescence experiments. This technique has the advantage of informing us about the subcellular localization of TAZ and ZEB1 and allows a semi-quantitative measure of TAZ and ZEB1 expression (gray level intensity determined using FIJI software on digital images) in the nucleus (like the immunoblotting experiment). We quantified TAZ and ZEB1 nuclear intensity staining by delimiting nuclei from DAPI staining and then quantifying the gray level mean intensity in the nucleus area, corresponding to TAZ and ZEB1 labelling (Figure 2B). This was done on more than 100 nuclei per condition, in three independent experiments. Moreover, the immunofluorescence has the second advantage of allowing the case by case study of individual cells. We can evaluate the double TAZ- and ZEB1-positive nuclear staining in order to highlight their co-overexpression induced in response to H. pylori (white arrows, Figure 2A). This co-overexpression of TAZ and ZEB1, determined by immunofluorescence, confirmed those observed by immunohistochemistry on human and mouse gastric tissue sections in Figure 3. This co-expression determined at single cell level by co-immunofluorescence staining cannot be provided by western blot experiments. In order to facilitate the visualization of nuclear staining of interest we added the DAPI staining in each immunofluorescence panels. (new Figures 2A, 4B, and new supplementary Figure S2C).

  1. Authors should show the image of invasion induced H. pylori (wild type and mutant).

- According to your suggestion, we have added representative images of the lower part of the microporous filter of type I-collagen-coated Transwell® inserts through which cells had invaded. Cells were stained with DAPI and counted in 5 different localisations of the Transwell® inserts. Invading cells after 7.13 WT H. pylori infection with or without siTAZ transfection are presented in new Figure S2-D and corresponding figure legend; line 416.

  1. Authors should measure the protein level of EMT markers in this study. Protein has the real function in regulating cell physiology, responses, behaviors et al.

- ZEB1 protein expression was evaluated not only at the mRNA level by RTqPCR but also at the protein level by immunofluorescence imaging. Concerning other EMT markers, as we previously published [1,3,4], we performed RTqPCR assays to study Vimentin, SNAI1 and MMP9 EMT markers in response to WT and mutant strains H. pylori infection or after TAZ knock-down. We agree in the relevance of protein study in physiological process, but we previously showed that the mRNA expression and protein expression (determined mainly by immunofluorescence and immunohistochemistry) was correlated in these models for ZEB1, Snail (SNAI1), Vimentin and MMP9 [1,3,4]. The best EMT marker was ZEB1, that is why we analysed ZEB1 in this study by three complementary techniques, ie immunofluorescence, RTqPCR and immunohistochemistry. In addition, properties associated to EMT, which reflect the functional consequences of these EMT-associated proteins, were confirmed by invasion and tumoursphere functional assays. We also added new data sets of SNAI1 in GC07 and MMP9 in MKN45 to strengthen these results (Figure 4A, lines 437-440, 433). Altogether, these results demonstrate an H. pylori-mediated TAZ implication in ZEB1 upregulation, EMT process and function.

- References cited in the responses to the reviewers comments:

  1. Molina-Castro, S.E.; Tiffon, C.; Giraud, J.; Boeuf, H.; Sifre, E.; Giese, A.; Belleannée, G.; Lehours, P.; Bessède, E.; Mégraud, F.; et al. The Hippo Kinase LATS2 Controls Helicobacter pylori-Induced Epithelial-Mesenchymal Transition and Intestinal Metaplasia in Gastric Mucosa. Cell Mol Gastroenterol Hepatol 2020, 9, 257–276, doi:10.1016/j.jcmgh.2019.10.007.
  2. Schindelin, J.; Arganda-Carreras, I.; Frise, E.; Kaynig, V.; Longair, M.; Pietzsch, T.; Preibisch, S.; Rueden, C.; Saalfeld, S.; Schmid, B.; et al. Fiji: an open-source platform for biological-image analysis. Nat. Methods 2012, 9, 676–682, doi:10.1038/nmeth.2019.
  3. Baud, J.; Varon, C.; Chabas, S.; Chambonnier, L.; Darfeuille, F.; Staedel, C. Helicobacter pylori Initiates a Mesenchymal Transition through ZEB1 in Gastric Epithelial Cells. PLoS ONE 2013, 8, e60315, doi:10.1371/journal.pone.0060315.
  4. Bessède, E.; Staedel, C.; Acuña Amador, L.A.; Nguyen, P.H.; Chambonnier, L.; Hatakeyama, M.; Belleannée, G.; Mégraud, F.; Varon, C. Helicobacter pylori generates cells with cancer stem cell properties via epithelial-mesenchymal transition-like changes. Oncogene 2014, 33, 4123–4131, doi:10.1038/onc.2013.380.
  5. Oda, T.; Kanai, Y.; Oyama, T.; Yoshiura, K.; Shimoyama, Y.; Birchmeier, W.; Sugimura, T.; Hirohashi, S. E-cadherin gene mutations in human gastric carcinoma cell lines. Proc. Natl. Acad. Sci. U.S.A. 1994, 91, 1858–1862, doi:10.1073/pnas.91.5.1858.
  6. Oliveira, M.J.; Costa, A.M.; Costa, A.C.; Ferreira, R.M.; Sampaio, P.; Machado, J.C.; Seruca, R.; Mareel, M.; Figueiredo, C. CagA associates with c-Met, E-cadherin, and p120-catenin in a multiproteic complex that suppresses Helicobacter pylori-induced cell-invasive phenotype. J. Infect. Dis. 2009, 200, 745–755, doi:10.1086/604727.
  7. Bessède, E.; Molina, S.; Acuña-Amador, L.; Dubus, P.; Staedel, C.; Chambonnier, L.; Buissonnière, A.; Sifré, E.; Giese, A.; Bénéjat, L.; et al. Deletion of IQGAP1 promotes Helicobacter pylori-induced gastric dysplasia in mice and acquisition of cancer stem cell properties in vitro. Oncotarget 2016, 7, 80688–80699, doi:10.18632/oncotarget.12486.
  8. Wroblewski, L.E.; Piazuelo, M.B.; Chaturvedi, R.; Schumacher, M.; Aihara, E.; Feng, R.; Noto, J.M.; Delgado, A.; Israel, D.A.; Zavros, Y.; et al. Helicobacter pylori targets cancer-associated apical-junctional constituents in gastroids and gastric epithelial cells. Gut 2015, 64, 720–730, doi:10.1136/gutjnl-2014-307650.
  9. Simeone, P.; Trerotola, M.; Franck, J.; Cardon, T.; Marchisio, M.; Fournier, I.; Salzet, M.; Maffia, M.; Vergara, D. The multiverse nature of epithelial to mesenchymal transition. Semin. Cancer Biol. 2019, 58, 1–10, doi:10.1016/j.semcancer.2018.11.004.
  10. Vergara, D.; Simeone, P.; Damato, M.; Maffia, M.; Lanuti, P.; Trerotola, M. The Cancer Microbiota: EMT and Inflammation as Shared Molecular Mechanisms Associated with Plasticity and Progression. J Oncol 2019, 2019, 1253727, doi:10.1155/2019/1253727.
  11. Vergara, D.; Ravaioli, S.; Fonzi, E.; Adamo, L.; Damato, M.; Bravaccini, S.; Pirini, F.; Gaballo, A.; Barbano, R.; Pasculli, B.; et al. Carbonic Anhydrase XII Expression Is Modulated during Epithelial Mesenchymal Transition and Regulated through Protein Kinase C Signaling. Int J Mol Sci 2020, 21, doi:10.3390/ijms21030715.
  12. Posselt, G.; Wiesauer, M.; Chichirau, B.E.; Engler, D.; Krisch, L.M.; Gadermaier, G.; Briza, P.; Schneider, S.; Boccellato, F.; Meyer, T.F.; et al. Helicobacter pylori-controlled c-Abl localization promotes cell migration and limits apoptosis. Cell Commun. Signal 2019, 17, 10, doi:10.1186/s12964-019-0323-9.
  13. Jang, E.J.; Jeong, H.; Han, K.H.; Kwon, H.M.; Hong, J.-H.; Hwang, E.S. TAZ suppresses NFAT5 activity through tyrosine phosphorylation. Mol. Cell. Biol. 2012, 32, 4925–4932, doi:10.1128/MCB.00392-12.
  14. Amieva, M.R.; Vogelmann, R.; Covacci, A.; Tompkins, L.S.; Nelson, W.J.; Falkow, S. Disruption of the epithelial apical-junctional complex by Helicobacter pylori CagA. Science 2003, 300, 1430–1434, doi:10.1126/science.1081919.
  15. Remue, E.; Meerschaert, K.; Oka, T.; Boucherie, C.; Vandekerckhove, J.; Sudol, M.; Gettemans, J. TAZ interacts with zonula occludens-1 and -2 proteins in a PDZ-1 dependent manner. FEBS Lett. 2010, 584, 4175–4180, doi:10.1016/j.febslet.2010.09.020.
  16. Nagy, T.A.; Frey, M.R.; Yan, F.; Israel, D.A.; Polk, D.B.; Peek, R.M. Helicobacter pylori regulates cellular migration and apoptosis by activation of phosphatidylinositol 3-kinase signaling. J. Infect. Dis. 2009, 199, 641–651, doi:10.1086/596660.
  17. Nagy, T.A.; Wroblewski, L.E.; Wang, D.; Piazuelo, M.B.; Delgado, A.; Romero-Gallo, J.; Noto, J.; Israel, D.A.; Ogden, S.R.; Correa, P.; et al. β-Catenin and p120 mediate PPARδ-dependent proliferation induced by Helicobacter pylori in human and rodent epithelia. Gastroenterology 2011, 141, 553–564, doi:10.1053/j.gastro.2011.05.004.
  18. Murata-Kamiya, N.; Kurashima, Y.; Teishikata, Y.; Yamahashi, Y.; Saito, Y.; Higashi, H.; Aburatani, H.; Akiyama, T.; Peek, R.M.; Azuma, T.; et al. Helicobacter pylori CagA interacts with E-cadherin and deregulates the beta-catenin signal that promotes intestinal transdifferentiation in gastric epithelial cells. Oncogene 2007, 26, 4617–4626, doi:10.1038/sj.onc.1210251.
  19. Franco, A.T.; Israel, D.A.; Washington, M.K.; Krishna, U.; Fox, J.G.; Rogers, A.B.; Neish, A.S.; Collier-Hyams, L.; Perez-Perez, G.I.; Hatakeyama, M.; et al. Activation of beta-catenin by carcinogenic Helicobacter pylori. Proc. Natl. Acad. Sci. U.S.A. 2005, 102, 10646–10651, doi:10.1073/pnas.0504927102.
  20. Azzolin, L.; Panciera, T.; Soligo, S.; Enzo, E.; Bicciato, S.; Dupont, S.; Bresolin, S.; Frasson, C.; Basso, G.; Guzzardo, V.; et al. YAP/TAZ incorporation in the β-catenin destruction complex orchestrates the Wnt response. Cell 2014, 158, 157–170, doi:10.1016/j.cell.2014.06.013.
  21. Ohnishi, N.; Yuasa, H.; Tanaka, S.; Sawa, H.; Miura, M.; Matsui, A.; Higashi, H.; Musashi, M.; Iwabuchi, K.; Suzuki, M.; et al. Transgenic expression of Helicobacter pylori CagA induces gastrointestinal and hematopoietic neoplasms in mouse. Proc. Natl. Acad. Sci. U.S.A. 2008, 105, 1003–1008, doi:10.1073/pnas.0711183105.
  22. Neal, J.T.; Peterson, T.S.; Kent, M.L.; Guillemin, K. H. pylori virulence factor CagA increases intestinal cell proliferation by Wnt pathway activation in a transgenic zebrafish model. Dis Model Mech 2013, 6, 802–810, doi:10.1242/dmm.011163.
  23. Varon, C.; Dubus, P.; Mazurier, F.; Asencio, C.; Chambonnier, L.; Ferrand, J.; Giese, A.; Senant-Dugot, N.; Carlotti, M.; Mégraud, F. Helicobacter pylori infection recruits bone marrow-derived cells that participate in gastric preneoplasia in mice. Gastroenterology 2012, 142, 281–291, doi:10.1053/j.gastro.2011.10.036.
  24. Belair, C.; Baud, J.; Chabas, S.; Sharma, C.M.; Vogel, J.; Staedel, C.; Darfeuille, F. Helicobacter pylori interferes with an embryonic stem cell micro RNA cluster to block cell cycle progression. Silence 2011, 2, 7, doi:10.1186/1758-907X-2-7.

Round 2

Reviewer 1 Report

I appreciate the authors` point-by-point response to all the questions raised in the first revision.

Reviewer 3 Report

Dear Editor,

The manuscript entitled “TAZ Controls Helicobacter pylori-Induced Epithelial-2 Mesenchymal Transition and Cancer Stem Cell-Like nvasive and Tumourigenic Properties” describes that the role of TAZ in the early steps of H. pylori-mediated gastric carcinogenesis. TAZ implication in EMT, invasion, and CSC-related tumourigenic properties are evaluated in three gastric epithelial cell lines infected by H. pylori. Authors show that H. pylori infection increases TAZ nuclear expression and TEADs transcriptional activity in a CagA-dependent manner. Nuclear TAZ and ZEB1 are co-overexpressed in cells harbouring a mesenchymal phenotype in vitro, and in areas of regenerative hyperplasia in gastric mucosa of H. pylori-infected patients and experimentally-infected mice, as well as at the invasive front of gastric carcinoma. TAZ silencing reduces ZEB1 expression, EMT phenotype and strongly inhibits invasion and tumoursphere formation induced by H. pylori. In conclusion, TAZ activation in response to H. pylori infection contributes to H. pylori-induced EMT, invasion, and CSC-like tumourigenic properties. TAZ overexpression in H. pylori-induced pre-neoplastic lesions and in GC could therefore constitute a biomarker of early transformation in gastric carcinogenesis.